# Ca²⁺ transients on the T cell surface trigger rapid integrin activation in a timescale of seconds

Yue Li[1,6], ShiHui Wang[1,6], YouHua Zhang[2,6], ZhaoYuan Liu[1], YunZhe Zheng[1], Kun Zhang[1], ShiYang Chen[3], XiaoYing Lv[4], MengWen Huang[3], XingChao Pan[1], YaJuan Zheng[1], MengYa Yuan[1], GaoXiang Ge [1,3], Yi Arial Zeng [1,3], ChangDong Lin [4,5,6] ✉ & JianFeng Chen [1,3] ✉

One question in lymphocyte homing is how integrins are rapidly activated to enable immediate arrest of fast rolling lymphocytes upon encountering chemokines at target vascular beds given the slow chemokine-induced integrin inside-out activation. Herein we demonstrate that chemokine CCL25-triggered Ca²⁺ influx induces T cell membrane-proximal external Ca²⁺ concentration ($[Ca^{2+}]_{ex}$) drop in 6 s from physiological concentration 1.2 mM to 0.3 mM, a critical extracellular Ca²⁺ threshold for inducing αLβ2 activation, triggering rapid αLβ2 activation and T cell arrest before occurrence of αLβ2 inside-out activation. Talin knockdown inhibits the slow inside-out activation of αLβ2 but not $[Ca^{2+}]_{ex}$ drop-triggered αLβ2 quick activation. Blocking Ca²⁺ influx significantly suppresses T cell rolling-to-arrest transition and homing to skin lesions in a mouse psoriasis model, thus alleviating skin inflammation. $[Ca^{2+}]_{ex}$ decrease-triggered rapid integrin activation bridges the gap between initial chemokine stimulation and slow integrin inside-out activation, ensuring immediate lymphocyte arrest and subsequent diapedesis on the right location.

The homing of lymphocytes from the bloodstream to lymphoid organs and inflamed tissues is essential to immune surveillance and host defense[1,2]. This process involves a highly ordered adhesion cascade, including selectin-mediated tethering and rolling of lymphocytes along the walls of high endothelial venules (HEVs), followed by integrin-mediated firm arrest and transendothelial migration of lymphocytes[3,4]. The correct location of lymphocyte extravasation at a specific target site is determined by the transition from rolling to firm arrest, which is triggered by the activation of integrins, such as αLβ2, upon chemokine stimulation[5]. Chemokines are expressed at high levels at lymphocyte homing target sites, inducing integrin activation that facilitates lymphocyte arrest on and trans-migration across the endothelium under both physiological and pathological conditions[3,6]. The binding of chemokines to their receptors on lymphocytes induces the activation of intracellular signaling and then promotes talin binding to the integrin β subunit cytoplasmic domain, which triggers integrin activation via inside-out signaling[7,8]. Notably, the classical inside-out activation of integrin by chemokines needs several minutes[9]. Considering lymphocytes roll in postcapillary venules at a relatively high speed ranging from 30–100 μm/s and the slow integrin

[1]State Key Laboratory of Multi-Cell Systems, Shanghai Institute of Biochemistry and Cell Biology, Center for Excellence in Molecular Cell Science, Chinese Academy of Sciences, Shanghai, China. [2]Department of Pathology, Shanghai Tenth People's Hospital, School of Medicine, Tongji University, Shanghai, China. [3]Key Laboratory of Systems Health Science of Zhejiang Province, School of Life Science, Hangzhou Institute for Advanced Study, University of Chinese Academy of Sciences, Hangzhou, China. [4]Fundamental Research Center, Shanghai YangZhi Rehabilitation Hospital (Shanghai Sunshine Rehabilitation Center), School of Life Sciences and Technology, Tongji University, Shanghai, China. [5]Frontier Science Center for Stem Cell Research, Tongji University, Shanghai, China. [6]These authors contributed equally: Yue Li, ShiHui Wang, YouHua Zhang, ChangDong Lin. ✉e-mail: linchangdong@tongji.edu.cn; jfchen@sibcb.ac.cn

activation by chemokine-triggered inside-out signaling[10,11], a quicker integrin activation is required to mediate the rapid arrest of rolling lymphocytes.

Integrins are metalloproteins and their functions are strictly dependent on and regulated by free $Ca^{2+}$ and $Mg^{2+}$ that physiologically exist in serum at millimolar-level[12,13]. Previous in vitro studies have revealed that $Ca^{2+}$ keeps integrins in an inactive state via binding to a metal ion binding site named ADMIDAS in the integrin β subunit I (βI) domain, and removal of extracellular $Ca^{2+}$ induces integrin quick activation in seconds[14,15]. Although the quick activation of integrin by removal of extracellular $Ca^{2+}$ has been reported for decades, no evidence shows that this mechanism can work in vivo because blood $Ca^{2+}$ concentration is relatively stable. Notably, chemokines can induce rapid and robust $Ca^{2+}$ flux, which makes us speculate that $Ca^{2+}$ flux may result in a transient $Ca^{2+}$ drop in the membrane-proximal external region of lymphocytes and subsequently induce quick activation of integrins.

In this study, we generate two biosensors to monitor T cell membrane-proximal external $Ca^{2+}$ concentration ($[Ca^{2+}]_{ex}$) and integrin activation in real time. A mouse model is established to express these biosensors in T cells simultaneously. Using both the isolated T cells and intravital imaging, we demonstrate that CCL25-induced $Ca^{2+}$ influx leads to a rapid decrease of $[Ca^{2+}]_{ex}$ on T cell surface to 0.3 mM, a critical $Ca^{2+}$ threshold for inducing integrin αLβ2 activation, in 6 s, and further to the lowest level 0.09 mM within 28 s, which induces the quick activation of αLβ2 and immediate arrest of T cells. The conventional chemokine-induced inside-out activation of integrins occurs after the rapid integrin activation by $[Ca^{2+}]_{ex}$ drop. Our findings demonstrate that second timescale activation of integrins can be achieved via chemokine-induced $Ca^{2+}$ transients on T cell surface, which update the current understanding of lymphocyte homing cascade by filling the gap between initial chemokine stimulation and slow inside-out activation of integrins. This mechanism enables lymphocytes to promptly stick to the spot where they encounter chemokines, ensuring the precise homing of lymphocytes to the target sites.

## Results

### Low extracellular $Ca^{2+}$ condition induces quick activation of integrin αLβ2

To investigate the effect of extracellular $Ca^{2+}$ change on integrin αLβ2 function in T cells, we isolated T cells from C57BL/6 J mouse spleen and examined T cell adhesion to the immobilized intercellular cell adhesion molecule-1 (ICAM-1) underflow condition in buffer containing 0.6 mM $Mg^{2+}$ and a series of concentrations of $Ca^{2+}$. The physiological concentrations of free $Ca^{2+}$ and $Mg^{2+}$ in serum (1.2 mM $Ca^{2+}$ and 0.6 mM $Mg^{2+}$) were set as control[16,17]. The firmly adherent cell number showed a significant increase when $Ca^{2+}$ decreased to 0.3 mM, indicating the activation of αLβ2 (Fig. 1a). Cell adhesion was enhanced gradually along with the further decrease in $Ca^{2+}$ concentration. Similarly, a decrease of $Ca^{2+}$ concentration to 0.3 mM and below significantly increased the binding of soluble ICAM-1 to T cells, suggesting an increased αLβ2 affinity for ICAM-1 (Fig. 1b). These data indicate that the ligand binding affinity and adhesiveness of integrin αLβ2 are significantly enhanced when extracellular $Ca^{2+}$ decreases to 0.3 mM and below.

Separation of integrin α/β subunit cytoplasmic tails is a critical conformational rearrangement during integrin activation[18]. To further investigate the effect of extracellular $Ca^{2+}$ decrease on αLβ2 activation, we used quantitative ratiometric fluorescence resonance energy transfer (FRET) imaging to assess the distance between the cytoplasmic domains of αL and β2 subunits[19–21]. Clover was fused to the C-terminus of αL subunit as the FRET donor, and mRuby2 was fused to the C-terminus of β2 subunit as acceptor (Fig. 1c). To express αL$_{Clover}$ and β2$_{mRuby2}$ simultaneously in mouse T cells, we generated conditional knockin alleles, *Itgal-loxP-Stop-loxP-Clover*

(*Itgal-LSL-Clover*) and *Itgb2-loxP-Stop-loxP-mRuby2* (*Itgb2-LSL-mRuby2*), with two *loxP* sites flanking the stop codon of *Itgal* or *Itgb2* and in front of *Clover* or *mRuby2* (Supplementary Fig. 1a, b). *Itgal-LSL-Clover* and *Itgb2-LSL-mRuby2* mice were crossed to generate *Itgal-LSL-Clover;Itgb2-LSL-mRuby2* mice. The mice were then crossed with *CD4-Cre* transgenic mice to generate *Itgal-LSL-Clover;Itgb2-LSL-mRuby2;CD4-Cre* mice carrying T cells expressing αL$_{Clover}$ and β2$_{mRuby2}$ simultaneously (Fig. 1c and Supplementary Fig. 1c). These mice developed normally. T cells were isolated from their spleens and seeded on ICAM-1 coated surface in buffer containing 0.6 mM $Mg^{2+}$ plus different concentrations of $Ca^{2+}$. Confocal images were acquired, and the ratio of acceptor/donor mean fluorescence intensity ($F_{mRuby2}/F_{Clover}$) was quantified. The images clearly showed a gradual decrease in $F_{mRuby2}/F_{Clover}$ along with the decrease of $Ca^{2+}$, indicating the gradual separation of αL/β2 tails and integrin activation (Fig. 1d). $F_{mRuby2}/F_{Clover}$ started to decrease when $Ca^{2+}$ dropped to 0.4 mM and showed a strong decrease in 0.3 mM $Ca^{2+}$ and below (Fig. 1e), which is in agreement with the enhanced cell adhesion and ICAM-1 binding in similar levels of $Ca^{2+}$ (Fig. 1a, b). In addition, chelation of $Ca^{2+}$ in solution using 5 mM Ethylene Glycol Tetraacetic Acid (EGTA) induced an immediate decrease of $F_{mRuby2}/F_{Clover}$ within seconds, indicating that the removal of extracellular $Ca^{2+}$ can induce rapid activation of αLβ2 (Fig. 1f).

### Establishing a cell membrane-anchored external CEPIA to monitor cell surface $Ca^{2+}$ dynamics

To monitor $[Ca^{2+}]_{ex}$ alteration on T cell surface in real-time, we established a plasma membrane-anchored external $Ca^{2+}$ biosensor by deleting the ER-targeting sequence from GEM-CEPIA1*er*[22] and adding PDGFR transmembrane (PDGFR-TM) sequence to the C-terminus of calcium-measuring organelle-entrapped protein indicator (CEPIA), which was composed of calmodulin (CaM), calmodulin-binding M13 skeletal muscle myosin light chain kinase (skMLCK) peptide, and a circularly permuted green fluorescent protein (cpGFP) allowing the indicator to turn into distinct conformations in different $Ca^{2+}$ concentrations (Fig. 2a). In order to express the membrane-anchored external CEPIA (CEPIA*external*) and αL$_{Clover}$β2$_{mRuby2}$ simultaneously in mouse T cells, we generated transgenic mice *Rosa26-loxP-Stop-loxP-CEPIAexternal* (*R26-LSL-CEPIAexternal*) (Supplementary Fig. 2a) and crossed with *Itgal-LSL-Clover;Itgb2-LSL-mRuby2;CD4-Cre* mice to obtain *R26-LSL-CEPIAexternal;Itgal-LSL-Clover;Itgb2-LSL-mRuby2;CD4-Cre* mice (Fig. 2a). Confocal imaging and Western blot of the membrane fraction of T cells from the mice confirmed that CEPIA*external* was expressed and displayed on T cell plasma membrane successfully (Fig. 2b, c and Supplementary Fig. 2b). Super-resolution microscopy imaging showed that CEPIA*external* biosensors were displayed on the extracellular side of plasma membrane (Fig. 2d). Co-expression of αL$_{Clover}$β2$_{mRuby2}$ with CEPIA*external* in T cells did not change CEPIA*external* Em510 lifetime, suggesting there is no energy transfer between CEPIA*external* and αL$_{Clover}$β2$_{mRuby2}$ (Supplementary Fig. 2c). The $Ca^{2+}$ calibration curve of T cell membrane-anchored CEPIA*external* ratio ($F_{Em450}/F_{Em510}$) showed a sensitive response in a range of $Ca^{2+}$ concentrations from 0.05 mM to 3.6 mM (Fig. 2e, f). Using this system, $[Ca^{2+}]_{ex}$ alteration on the T cell surface could be monitored efficiently in real-time.

### Ionomycin induced-$Ca^{2+}$ influx triggers $[Ca^{2+}]_{ex}$ drop and αLβ2 quick activation

To investigate whether $Ca^{2+}$ influx can decrease $[Ca^{2+}]_{ex}$ to a level that can trigger αLβ2 activation, we treated cells with ionomycin, a potent calcium ionophore, and monitored $[Ca^{2+}]_{ex}$ change in real time. Upon ionomycin addition, the T cell membrane-anchored CEPIA*external* ratio showed a rapid decrease in seconds (Fig. 3a, b and Supplementary Fig. 3a), indicating a rapid $[Ca^{2+}]_{ex}$ decrease on the T cell surface. $[Ca^{2+}]_{ex}$ decreased to 0.3 mM, the critical threshold for inducing αLβ2

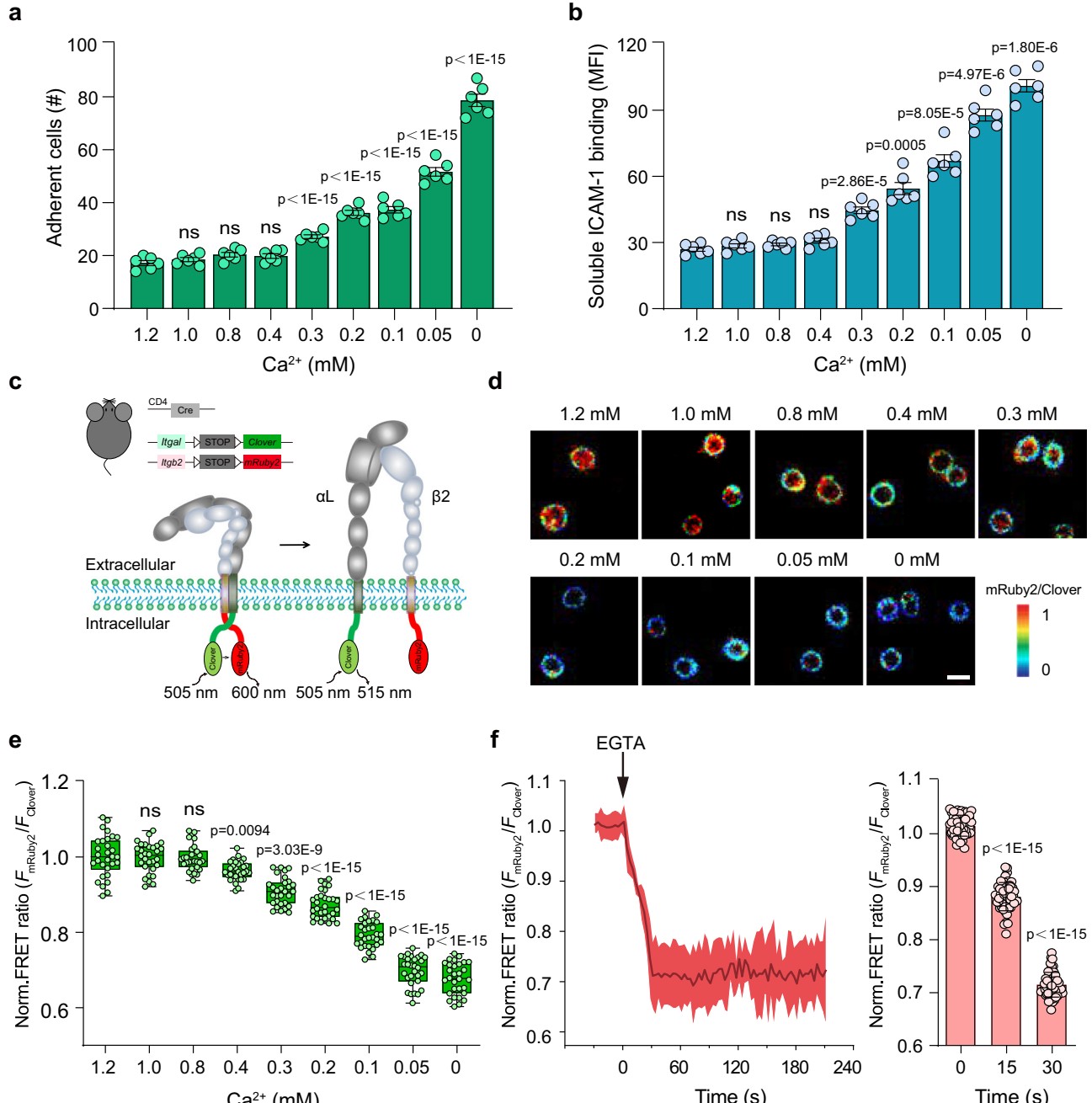

**Fig. 1 | Low extracellular Ca²⁺ condition induces rapid activation of integrin αLβ2.** Mouse splenic T cells were suspended in a buffer containing 0.6 mM Mg²⁺ plus the indicated concentration of Ca²⁺. 0 mM [Ca²⁺] means a Ca²⁺ concentration below the detection limit (2.34 μM Ca²⁺) of the Calcium Quantitation Kit (36361, AAT Bioquest). **a** ICAM-1 (20 μg/ml) was immobilized on petri dishes. Adhesion of T cells to the immobilized ICAM-1 substrates at a wall shear stress of 1 dyn/cm² was examined (*n* = 6). **b** Binding of soluble ICAM-1 to T cells was calculated with the specific mean fluorescence intensity (MFI) (*n* = 6). **c** Schematic diagrams of the experimental setup for integrin αLβ2 tail FRET system (αL_Cloverβ2_mRuby2) to monitor the separation of αLβ2 cytoplasmic domains (bottom) and the strategy to generate *Itgal-LSL-Clover;Itgb2-LSL-mRuby2;CD4-Cre* mice bearing T cells expressing αL_Cloverβ2_mRuby2 (top). **d** Representative pseudocolored αLβ2 tail FRET ratio (*F*_mRuby2/*F*_Clover) images of T cells expressing αL_Cloverβ2_mRuby2 on the immobilized ICAM-1 (20 μg/ml) substrates. Scale bar, 6 μm. Images are from one representative experiment out of three. **e** Quantification of αLβ2 tail FRET ratio.

The FRET ratio of each cell was normalized to the mean value of cells in 1.2 mM Ca²⁺. Data are presented as box-and-whisker plots showing the median (central line), 25th–75th percentile (bounds of the box), and 5th–95th percentile (whiskers) (*n* = 30 cells for each condition from 3 experiments). **f** Time course of αLβ2 tail FRET ratio change in T cells on the immobilized ICAM-1 (20 μg/ml) substrates upon chelation of Ca²⁺ with 5 mM EGTA in buffer containing 1.2 mM Ca²⁺ plus 0.6 mM Mg²⁺ (left). EGTA was added at time point 0. The FRET ratio change was normalized to the mean value of cells before EGTA treatment. The solid lines represent the mean; shaded areas, s.e.m. (*n* = 50 cells from 3 experiments). The statistic results at representative time points were shown (right). Data represent the mean ± s.e.m. in (**a**, **b** and **f**). ns, not significant (one-way ANOVA with Dunnett's test in (**a**), Brown-Forsythe and Welch one-way ANOVA with Dunnett's test in (**b**) and (**e**) to compare the means of different Ca²⁺ concentration groups to the mean of 1.2 mM Ca²⁺ group; one-way ANOVA with Dunnett's test in (**f**) to compare the means of 15 s and 30 s groups to the mean of 0 s group). Source data are provided as a Source Data file.

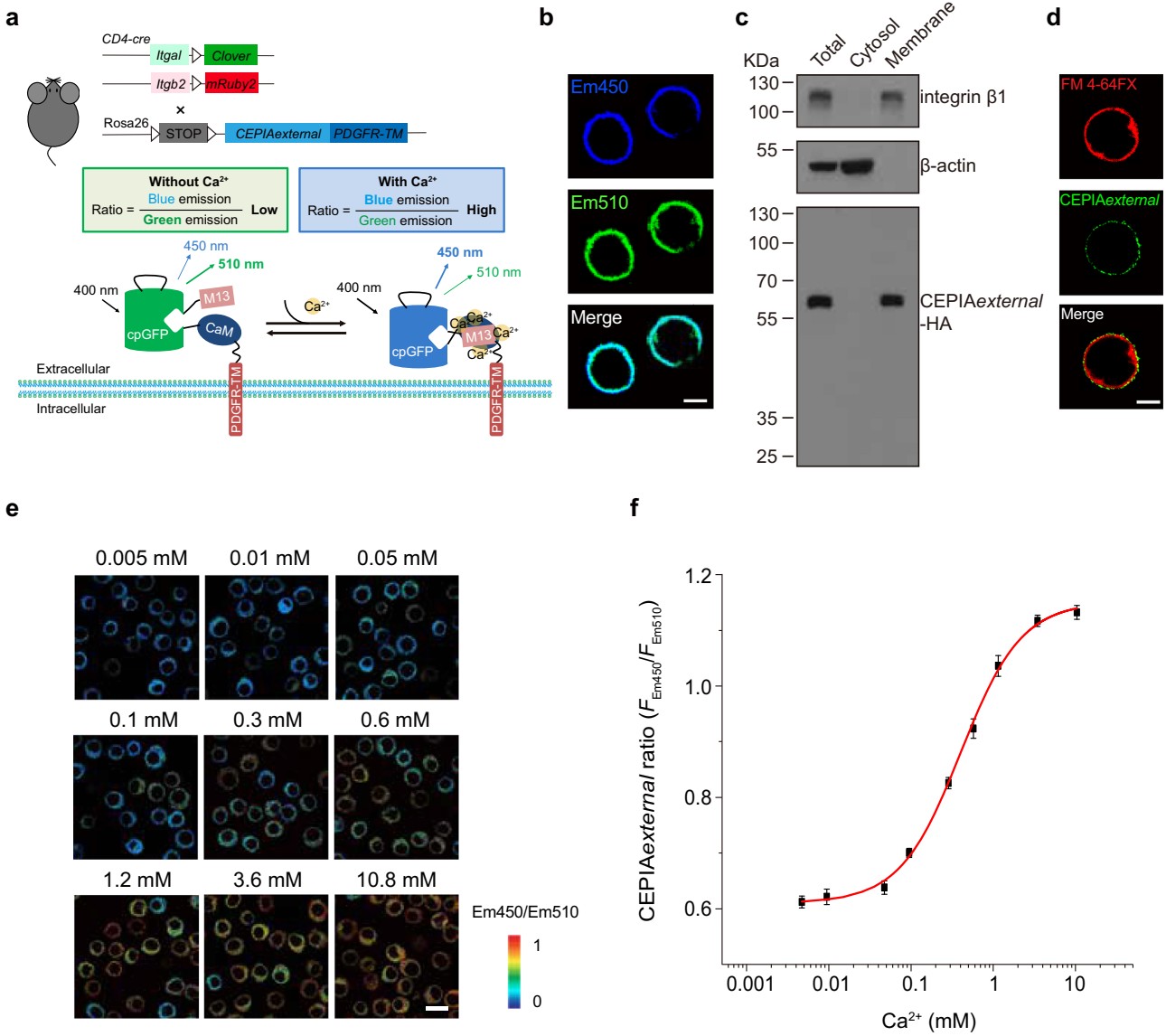

**Fig. 2 | Establishing a cell membrane-anchored CEPIA*external* to monitor T cell surface Ca²⁺ dynamics. a** Schematic diagrams of the experimental setup for the membrane-anchored CEPIA*external* (bottom) and the strategy to generate *R26-LSL-CEPIAexternal;Itgal-LSL-Clover;Itgb2-LSL-mRuby2;CD4-Cre* mice bearing T cells expressing both the CEPIA*external* and αL$_{Clover}$β2$_{mRuby2}$ (top). **b** Representative fluorescence images of mouse splenic T cells showing the distribution of CEPIA*external* on the plasma membrane in Em450 and Em510 channels. Scale bar, 3 μm. Images are from one representative experiment out of three. **c** Immunoblot analysis of CEPIA*external*, integrin β1, and β-actin in the whole-cell lysate (total), cytosol or membrane fraction of mouse splenic T cells. Images are from one representative experiment out of three. **d** Representative fluorescence images of mouse splenic T cells showing the distribution of CEPIA*external* and the plasma membrane indicated by FM 4-64FX. Scale bar, 3 μm. Images are from one representative experiment out of three. **e** Representative pseudocoloured $F_{Em450}/F_{Em510}$ images of mouse splenic T cells in the indicated extracellular Ca²⁺ concentrations. Scale bar, 6 μm. Images are from one representative experiment out of three. **f** Fitted curve of CEPIA*external* ratio ($F_{Em450}/F_{Em510}$) in response to a series of concentrations of Ca²⁺. $n = 60$ cells from 3 experiments, the bars represent mean ± s.e.m. Source data are provided as a Source Data file.

activation, in 6 s and reached the lowest level 0.08 mM in 28 s (Fig. 3c). Ionomycin-induced Ca²⁺ drop was limited to the cell membrane-proximal external region but did not alter the entire Ca²⁺ concentration in the solution (Fig. 3d). Besides using CEPIA*external* to measure Ca²⁺ concentration on T cell surface, we also used Rhod Red (Ca²⁺ fluorescence probe) to confirm the ionomycin-induced Ca²⁺ drop in T cell surface region (see below). Upon ionomycin addition, Rhod Red intensity in the region close to the external plasma membrane of T cells showed a rapid decrease and reached to the lowest level in about 28 s (see below), indicating a rapid [Ca²⁺]$_{ex}$ decrease on the T cell surface. These Ca²⁺ dynamics in T cell surface region showed by Rhod Red is consistent with those observed using membrane-anchored CEPIA (Fig. 3a–c).

In addition to inducing [Ca²⁺]$_{ex}$ decrease, Ca²⁺ influx can induce an increase of cytosolic Ca²⁺ concentration ([Ca²⁺]$_{cyto}$) (Supplementary Fig. 3b), which may promote integrin activation via inside-out signaling[23]. To study the net effect of [Ca²⁺]$_{ex}$ decrease on αLβ2 activation, we used BAPTA-AM to chelate the intracellular Ca²⁺ to eliminate ionomycin-induced [Ca²⁺]$_{cyto}$ increase in T cells (Supplementary Fig. 3b). BAPTA-AM treatment did not affect ionomycin-induced [Ca²⁺]$_{ex}$ decrease (Fig. 3e).

Next, we assessed αLβ2 activation in T cells in response to ionomycin-induced [Ca²⁺]$_{ex}$ decrease. Ionomycin treatment resulted in a rapid and continuous decrease in integrin tail FRET signal within 28 s, indicating a quick activation of αLβ2 (Fig. 3f, g). BAPTA-AM treatment did not affect the rapid integrin activation but specifically prevented

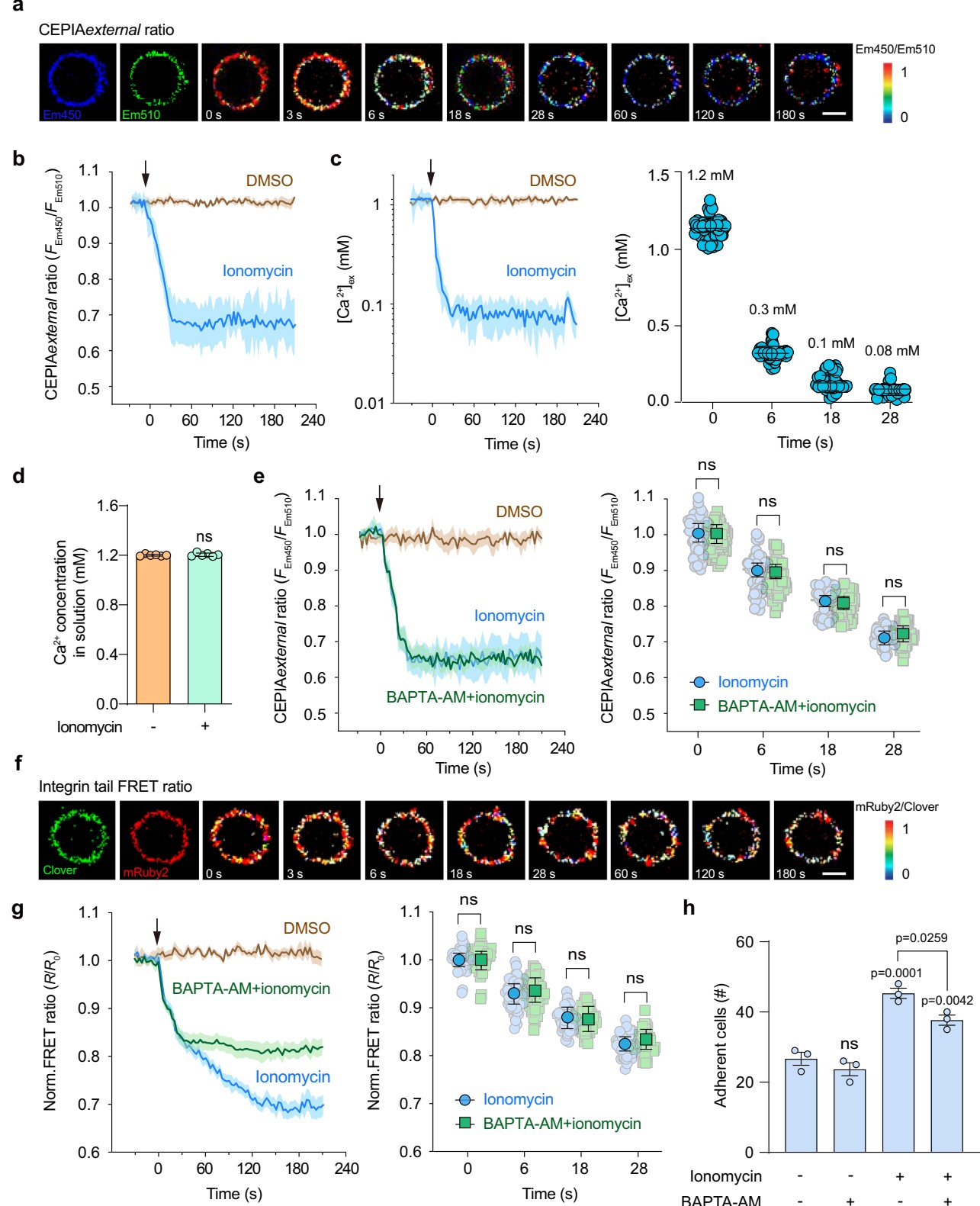

the decrease in FRET signal after 28 s (Fig. 3g), suggesting that intracellular $Ca^{2+}$-induced integrin activation via inside-out signaling occurred after 28 s, and could be inhibited by chelating intracellular $Ca^{2+}$. In line with these results, ionomycin enhanced T cell adhesion on ICAM-1 substrate in flow, which was partially inhibited by BAPTA-AM treatment (Fig. 3h).

**Ionomycin-induced persistent low $[Ca^{2+}]_{ex}$ is dependent on continuous $Ca^{2+}$ influx and slow $Ca^{2+}$ diffusion from solution to cell surface**

$Ca^{2+}$ concentration on the T cell surface remained low for at least 180 s after ionomycin administration (Fig. 3a–c), implying a continuous $Ca^{2+}$ influx. However, the intracellular cytosolic $Ca^{2+}$ concentration ($[Ca^{2+}]_{cyto}$)

**Fig. 3 | Ionomycin induces [Ca²⁺]ₑₓ drop and integrin αLβ2 activation on T cell surface.** Splenic T cells were isolated from *R26-LSL-CEPIAexternal;Itgal-LSL-Clover;Itgb2-LSL-mRuby2;CD4-Cre* mice and suspended in buffer containing 1.2 mM Ca²⁺ and 0.6 mM Mg²⁺. Ionomycin (final concentration 1 μM) was added at time point 0. **a** Representative pseudocolor image of CEPIA*external* ratio on the T cell surface in response to ionomycin stimulation. The first two images show the distribution of CEPIA*external* in Em450 and Em510 channels. Scale bar, 3 μm. Images are from one representative experiment out of three. **b** Time course of CEPIA*external* ratio change on the T cell surface in response to ionomycin stimulation (*n* = 30 cells from 3 experiments). **c** [Ca²⁺]ₑₓ curve was plotted by converting the CEPIA*external* ratio in (**b**) to Ca²⁺ concentration according to the Ca² calibration curve in Fig. 2f (left), and [Ca²⁺]ₑₓ values at the representative time points were shown (right) (*n* = 65 cells from 3 experiments). **d** Ca²⁺ concentration in the buffer was measured using a Calcium Colorimetric Assay Kit (S1063S, Beyotime) before and after ionomycin treatment (*n* = 6). **e** Time course of CEPIA*external* ratio change in T cells pretreated with 100 μM BAPTA-AM or DMSO vehicle control in response to ionomycin stimulation (left) and CEPIA*external* ratios at the representative time

points were shown (right) (*n* = 60 cells from 3 experiments). **f** Representative pseudocolor images of the αLβ2 tail FRET ratio on the T cell surface in response to ionomycin stimulation. The first two images show the Clover and mRuby2 signals. Scale bar, 3 μm. Images are from one representative experiment out of three. **g** Time course of αLβ2 tail FRET ratio change in T cells pretreated with 100 μM BAPTA-AM or DMSO vehicle control in response to ionomycin stimulation (left), and the normalized FRET ratios at the representative time points were shown (right) (*n* = 60 cells from 3 experiments). The FRET ratio is normalized to the mean value of cells before the addition of stimuli (*R/R₀*). **h** Effect of ionomycin treatment on the adhesion of T cells pretreated with 100 μM BAPTA-AM or DMSO vehicle control to the immobilized ICAM-1 (20 μg/ml) substrates at a wall shear stress of 1 dyn/cm² (*n* = 3). The solid lines represent the mean; shaded areas, s.e.m. in (**b**, **c**, **e**, and **g**). Data represent the mean ± s.e.m. in (**c**, **d**, **e**, **g** and **h**). ns, not significant (unpaired two-tailed Student's *t* test and unpaired two-tailed Welch's unequal variance *t* test in (**d**, **e**, and **g**); one-way ANOVA with Dunnett's test in (**h**). Source data are provided as a Source Data file.

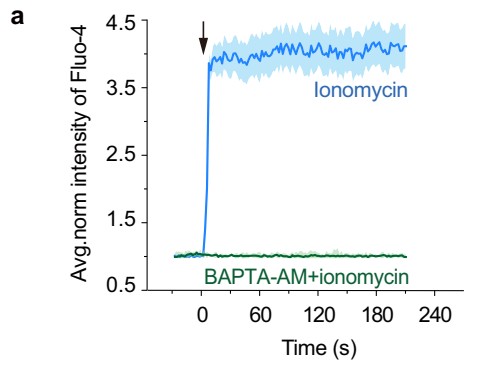

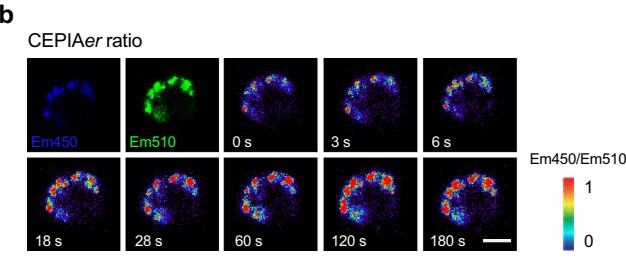

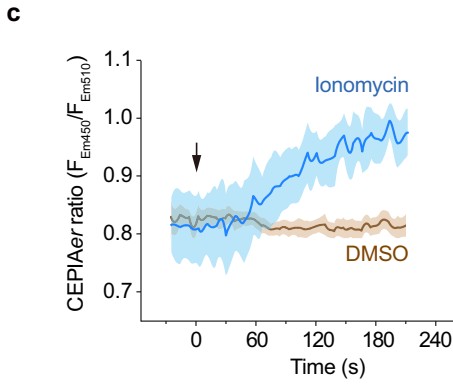

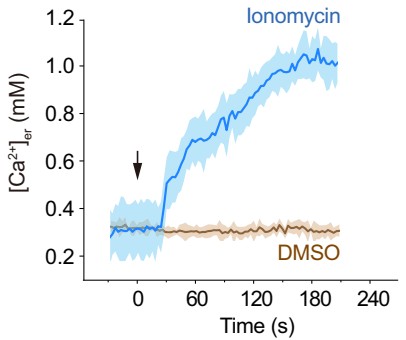

**Fig. 4 | Effects of ionomycin treatment on intracellular Ca²⁺ dynamics in T cells.** Splenic T cells from WT mice transfected with CEPIA*er* were suspended in a buffer containing 1.2 mM Ca²⁺ and 0.6 mM Mg²⁺. Ionomycin (final concentration 1 μM) was added at time point 0. **a** Cytosolic Ca²⁺ ([Ca²⁺]ᵧₜₒ) change was detected by Fluo-4 in T cells pretreated with 100 μM BAPTA-AM or DMSO vehicle control in response to stimulation with 1 μM ionomycin. **b** Representative pseudocolor images of CEPIA*er* ratio in response to stimulation with 1 μM ionomycin. The first two images show the

distribution of CEPIA*er* in Em450 and Em510 channels. Scale bar, 3 μm. Images are from one representative experiment out of three. **c** Time course of CEPIA*er* ratio change (left panel) and the corresponding change of Ca²⁺ in ER ([Ca²⁺]ₑᵣ) (right panel) in response to stimulation with 1 μM ionomycin. The solid lines represent the mean; shaded areas, s.e.m. in (**a**) and (**c**). *n* = 12 cells from 3 experiments. Source data are provided as a Source Data file.

only showed a short-time increase and then remained stable (Fig. 4a), suggesting the inflowed Ca²⁺ was stored somewhere in the cells. Given the endoplasmic reticulum (ER)'s role as the main calcium reservoir[24], we hypothesize that ER stores the inflowed Ca²⁺ and, therefore, maintains a stable level of [Ca²⁺]cyto. Using ER-targeting CEPIA (CEPIA*er*)[22] imaging to monitor the level of Ca²⁺ in ER ([Ca²⁺]ₑᵣ), we found that [Ca²⁺]ₑᵣ increased continuously upon ionomycin addition (Fig. 4b, c). Thus, ionomycin induces a continuous Ca²⁺ influx for at least 180 s, which induces a short time increase in [Ca²⁺]cyto level and a continuous increase in [Ca²⁺]ₑᵣ.

Besides the continuous Ca²⁺ influx, the slow Ca²⁺ diffusion from solution to T cell surface might also contribute to the Ca²⁺ influx-induced persistent low level of [Ca²⁺]ₑₓ in a high Ca²⁺ containing environment. Most cells, including lymphocytes, are covered by a dense glycocalyx, which resides extracellularly on the cell membrane, surrounding the cell like a cloak[25,26]. This special structure contributes to the formation of a physical and charged barrier modulating the flow of low molecular substrates or ions into and out of the cells[27–29], suggesting the delay of Ca²⁺ diffusion from the solution to the external surface of T cells. Indeed, the addition of 1.2 mM Ca²⁺ into the solution

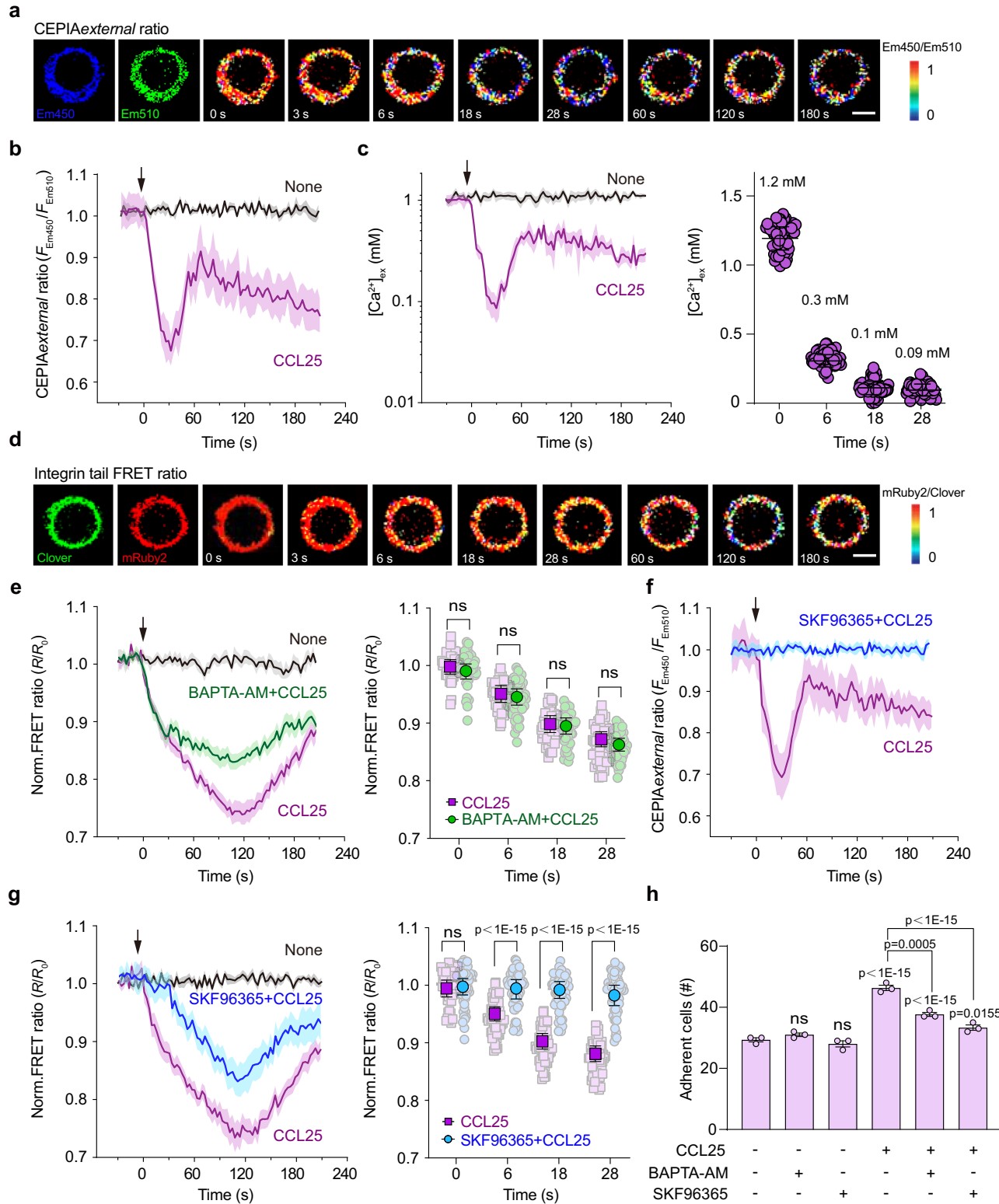

rapidly increased $[Ca^{2+}]_{ex}$ on the surface of CEPIA-labeled agarose beads (Supplementary Fig. 4) due to the free diffusion of $Ca^{2+}$. By contrast, $[Ca^{2+}]_{ex}$ on T cell surface showed slowly increase, indicating that $Ca^{2+}$ diffuses slowly from solution to T cell surface.

### Chemokine-triggered $Ca^{2+}$ influx induces $[Ca^{2+}]_{ex}$ drop and αLβ2 quick activation

CCL25 is an important chemokine for T cell homing and can induce rapid $Ca^{2+}$ influx through store-operated calcium entry (SOCE) and

calcium-release activated calcium (CRAC) channels when binds to its receptor CCR9[23,30]. CCL25 treatment induced a rapid decrease in CEPIA*external* ratio and $[Ca^{2+}]_{ex}$ on T cell surface from 1.2 mM to 0.3 mM in 6 s (Fig. 5a–c). $[Ca^{2+}]_{ex}$ further decreased to 0.1 mM in 18 s and reached its lowest level of 0.09 mM in 28 s, then recovered to about 0.6 mM in the following 30 s (Fig. 5c). In addition, similar $[Ca^{2+}]_{ex}$ drop patterns were observed on the surface of T cells upon CCL25 treatment when using Rhod Red to monitor extracellular $Ca^{2+}$ change (Fig. 6c, d). In line with these results, CCL25 treatment induced a rapid

**Fig. 5 | CCL25 induces $[Ca^{2+}]_{ex}$ drop and integrin αLβ2 activation on T cell surface.** Splenic T cells were isolated from *R26-LSL-CEPIAexternal;Itgal-LSL-Clover;Itgb2-LSL-mRuby2;CD4-Cre* mice and suspended in buffer containing 1.2 mM $Ca^{2+}$ and 0.6 mM $Mg^{2+}$. CCL25 (final concentration 0.5 μg/ml) were added at time point 0. **a** Representative pseudocolor images of CEPIA*external* ratio on the T cell surface in response to CCL25 treatment. The first two images show the distribution of CEPIA*external* in Em450 and Em510 channels. Scale bar, 3 μm. Images are from one representative experiment out of three. **b** Time course of CEPIA*external* ratio change on the T cell surface in response to CCL25 stimulation. Untreated T cells were used as control (None) (*n* = 30 cells from 3 experiments). **c** $[Ca^{2+}]_{ex}$ curve was plotted by converting the CEPIA*external* ratio in (**b**) to $Ca^{2+}$ concentration according to the calibration curve in Fig. 2f (left), and $[Ca^{2+}]_{ex}$ values at the representative time points were shown (right) (*n* = 65 cells from 3 experiments). **d** Representative pseudocolor images of the αLβ2 tail FRET ratio on the T cell surface in response to CCL25 stimulation. The first two images show the Clover and mRuby2 signals. Scale bar, 3 μm. Images are from one representative experiment out of three. **e** Time course of αLβ2 tail FRET ratio change in T cells pretreated with 100 μM BAPTA-AM or vehicle control in response to CCL25 stimulation and the

normalized FRET ratios at the representative time points were shown. The FRET ratio is normalized to the mean value of cells before the addition of stimuli ($R/R_0$) (*n* = 60 cells from 3 experiments). **f** Time course of CEPIA*external* ratio change in T cells pretreated with 100 μM SKF96365 or vehicle control in response to stimulation with 0.5 μg/ml CCL25 (*n* = 60 cells from 3 experiments). **g** Time course of αLβ2 tail FRET ratio change in T cells pretreated with 100 μM SKF96365 or vehicle control in response to CCL25 stimulation and the normalized FRET ratios at the representative time points were shown. The FRET ratio is normalized to the mean value of cells before the addition of stimuli ($R/R_0$) (*n* = 60 cells from 3 experiments). **h** Adhesion of T cells pretreated with 100 μM BAPTA-AM, 100 μM SKF96365, or vehicle control to the immobilized ICAM-1 (20 μg/ml) alone or ICAM-1 (20 μg/ml) plus CCL25 (2 μg/ml) substrates at a wall shear stress of 1 dyn/cm² (*n* = 3). The solid lines represent the mean; shaded areas, s.e.m. in (**b**, **c**, **e**, **f**, and **g**). Data represent the mean ± s.e.m. in (**c**, **e**, **g**, and **h**). ns, not significant (unpaired two-tailed Student's *t* test and unpaired two-tailed Welch's unequal variance *t* test in (**e**) and (**g**); one-way ANOVA with Dunnett's test in (**h**). Source data are provided as a Source Data file.

decrease in integrin tail FRET signal in 28 s, suggesting a quick activation of αLβ2 that was triggered by the decrease in $Ca^{2+}$ on T cell surface. During this period, αLβ2 activation was not affected by chelating intracellular $Ca^{2+}$ with BAPTA-AM treatment (Fig. 5d, e), indicating it is distinct from the conventional slow integrin inside-out activation.

Conventional slow integrin inside-out activation by chemokines includes intracellular $Ca^{2+}$-induced and intracellular $Ca^{2+}$-independent activation of integrins[31]. Chelating intracellular $Ca^{2+}$ with BAPTA-AM (Supplementary Fig. 5) partially prevented the decrease in integrin tail FRET signal after 28 s upon CCL25 treatment (Fig. 5e), suggesting BAPTA-AM inhibited the intracellular $Ca^{2+}$-induced slow integrin inside-out activation but did not affect intracellular $Ca^{2+}$-independent integrin inside-out activation. SKF96365 is a SOCE inhibitor and also blocks TRPC channels and voltage-gated $Ca^{2+}$ channels[32,33], which can potently obstruct chemokine-induced $Ca^{2+}$-influx. Blocking $Ca^{2+}$-influx with SKF96365 inhibited CCL25-induced $[Ca^{2+}]_{ex}$ decrease on T cell surface (Fig. 5f) and the associated αLβ2 quick activation in 28 s (Fig. 5g). The remained αLβ2 activation after 28 s should be due to the CCL25-induced $Ca^{2+}$-independent inside-out activation of integrin. Consistent with these results, the CCL25-enhanced T cell adhesion on ICAM-1 substrate in flow was partially inhibited by BAPTA-AM and SKF96365 (Fig. 5h). SKF96365 showed stronger inhibition than BAPTA-AM because SKF96365 inhibits both $[Ca^{2+}]_{ex}$ decrease-induced and intracellular $Ca^{2+}$-dependent activations of αLβ2.

To further dissect the effects of BAPTA-AM and SKF96365 on $Ca^{2+}$ influx-induced intracellular $Ca^{2+}$ dynamics, we also measured the local internal submembrane $Ca^{2+}$ ($[Ca^{2+}]_{in}$) changes by establishing a plasma membrane-anchored internal CEPIA (CEPIA*internal*) with N-terminal fused PDGFR-TM sequence (Supplementary Fig. 6a). Super-resolution microscopy imaging confirmed that CEPIA*internal* was located at the internal side of cell plasma membrane (Supplementary Fig. 6b). CCL25 treatment induced a rapid increase in CEPIA*internal* ratio and $[Ca^{2+}]_{in}$ in the first few seconds, then recovered back gradually (Supplementary Fig. 6c, d), which is similar to the results of CCL25-induced $[Ca^{2+}]_{cyto}$ change in the cytosol (Supplementary Fig. 5). Thus, CCL25-triggered $Ca^{2+}$ influx induces similar $Ca^{2+}$ increase patterns in both local internal submembrane region and cytosol. Blocking $Ca^{2+}$ influx with SKF96365 not only inhibited $[Ca^{2+}]_{cyto}$ and $[Ca^{2+}]_{in}$ increases as did by chelating intracellular $Ca^{2+}$ with BAPTA-AM (Supplementary Fig. 5 and Supplementary Fig. 6d), but also blocked CCL25-induced $[Ca^{2+}]_{ex}$ drop (Fig. 5f). The major difference between the effects of SKF96365 and BAPTA-AM is that SKF96365 but not BAPTA-AM can inhibit CCL25-induced $[Ca^{2+}]_{ex}$ drop (Fig. 5f and Supplementary Fig. 6e) and its associated αLβ2 quick activation in 28 s upon chemokine stimulation (Fig. 5g, e), suggesting αLβ2 quick activation is essentially caused by $Ca^{2+}$ influx-induced $[Ca^{2+}]_{ex}$ drop.

## $[Ca^{2+}]_{ex}$ decrease-triggered αLβ2 quick activation is independent of integrin inside-out signaling

Talin is an essential mediator of integrin activation through inside-out signaling[34,35]. Silencing of talin inhibits conventional inside-out activation of integrins[35,36]. Knockdown of talin in T cells (Supplementary Fig. 7) did not affect $[Ca^{2+}]_{ex}$ decrease-associated quick activation of αLβ2 within 28 s upon ionomycin or CCL25 treatment, only specifically inhibited the subsequent slow integrin inside-out activation (Fig. 7a, b), suggesting that the rapid activation of αLβ2 triggered by $[Ca^{2+}]_{ex}$ decrease is independent of conventional integrin inside-out activation signaling.

## $Ca^{2+}$ influx-induced $[Ca^{2+}]_{ex}$ drop and its associated αLβ2 quick activation is critical for T cell transition from rolling to arrest in vivo

Next, we investigated the role of $Ca^{2+}$ influx-induced $[Ca^{2+}]_{ex}$ drop and its associated αLβ2 quick activation in T cell homing in skin post-capillary venules in *R26-LSL-CEPIAexternal;Itgal-LSL-Clover;Itgb2-LSL-mRuby2;CD4-Cre* mice. Venules were visualized with dextran Texas Red and CEPIA*external* was clearly visible in T cells (Fig. 8a and Supplementary Movie 1). The transition of T cells from rolling to arrest was successfully captured in skin microcirculation venules (Fig. 8b and Supplementary Movie 1). During the T cell rolling-to-arrest transition, CEPIA*external* ratio decreased rapidly, suggesting the quick decrease of $[Ca^{2+}]_{ex}$ on the T cell surface (Fig. 8c, d and Supplementary Movie 2). The consequent decrease of integrin tail FRET indicated the activation of αLβ2 (Fig. 8c, e). T cell rolling velocity decreased from about 55 to 0 μm/s within 10 s in line with the decrease of $[Ca^{2+}]_{ex}$ from about 1.2 mM to 0.1 mM and the consequent αLβ2 activation (Fig. 8f). Notably, different from the recovery of $[Ca^{2+}]_{ex}$ in T cells 30 s post CCL25 stimulation in vitro (Fig. 5b, c), we did not observe $[Ca^{2+}]_{ex}$ recovery in the homing T cells 30 s post the initial decrease of CEPIA*external* ratio (Fig. 8d), which could be due to the effect of different chemokines. Indeed, CXCL9 and CXCL10 contribute to T cell recruitment to the skin[37], these chemokines induced constant $[Ca^{2+}]_{ex}$ drop at least within 210 s (Supplementary Fig. 8), which is different from the transient decrease of $[Ca^{2+}]_{ex}$ induced by CCL25.

## Blockade of $Ca^{2+}$ influx-induced $[Ca^{2+}]_{ex}$ drop alleviates IMQ-induced psoriasis

Inhibition of αLβ2-mediated T cell infiltration into the dermis is a promising therapeutic strategy for psoriasis[38]. We then used a well-characterized imiquimod (IMQ)-induced psoriasis mouse model to investigate the biological significance of $Ca^{2+}$ influx-induced $[Ca^{2+}]_{ex}$ drop and its associated αLβ2 quick activation in the pathology of psoriasis. We first examined the contribution of $Ca^{2+}$ influx-induced $[Ca^{2+}]_{ex}$ drop in T cell homing to inflamed skin tissue in psoriasis mice.

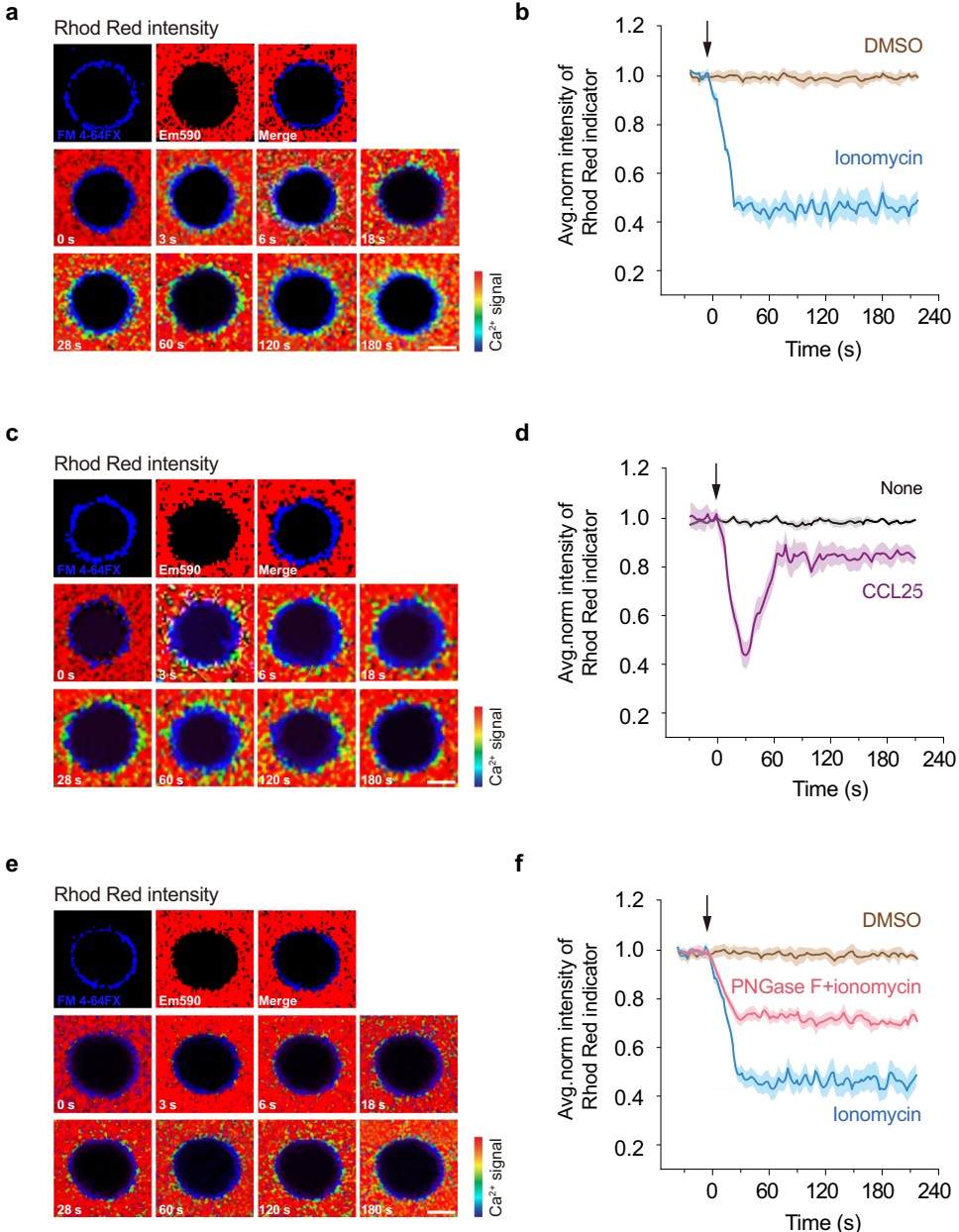

**Fig. 6 | Ionomycin and CCL25 induce $[Ca^{2+}]_{ex}$ drop on T cell surface using Rhod Red indicator.** Splenic T cells isolated from WT mice were suspended in a buffer containing 1.2 mM $Ca^{2+}$ and 0.6 mM $Mg^{2+}$, and 1× Rhod Red stock solution. Ionomycin (final concentration 1 μM) or CCL25 (final concentration 0.5 μg/ml) were added at time point 0. T cells were pretreated with 1000 units/ml PNGase F in (**e**) and (**f**). **a** Representative pseudocolour images of Rhod Red ratio in the solution in response to stimulation with 1 μM ionomycin. The plasma membrane was indicated by FM 4-64FX. Scale bar, 3 μm. Images are from one representative experiment out of three. **b** Time course of Rhod Red intensity in T cell surface region in response to stimulation with 1 μM ionomycin. **c** Representative pseudocolor images of Rhod Red ratio in the solution in response to stimulation with 0.5 μg/ml CCL25. The

plasma membrane was indicated by FM 4-64FX. Scale bar, 3 μm. Images are from one representative experiment out of three. **d** Time course of Rhod Red intensity in T cell surface region in response to stimulation with 0.5 μg/ml CCL25. **e** Representative pseudocolor images of Rhod Red ratio in the solution in response to stimulation with 1 μM ionomycin. The plasma membrane was indicated by FM 4-64FX. Scale bar, 3 μm. Images are from one representative experiment out of three. **f** Time course of Rhod Red intensity in PNGase F-treated T cell surface region in response to stimulation with 1 μM ionomycin. The solid lines represent the mean; shaded areas, s.e.m. in (**b**, **d**, and **f**). $n = 12$ cells from 3 experiments. Source data are provided as a Source Data file.

Cell-tracer 647-labeled T cells expressing CEPIA*external* and integrin $\alpha L_{Clover}\beta 2_{mRuby2}$ were pretreated with SKF96365 and then injected into the receipt psoriasis mice (Fig. 9a). Inhibition of $Ca^{2+}$ influx with SKF96365 pretreatment can last for 3 h[39], which is long enough for T cell homing experiment. Compared with DMSO-treated T cells, SKF96365 treatment significantly reduced the number of arrested T cells in inflamed skin venules (Fig. 9b, c and Supplementary Movie 3), suggesting that $Ca^{2+}$ influx-induced $[Ca^{2+}]_{ex}$ drop is critical for T cell

homing to inflamed skin. To further assess the role of $Ca^{2+}$ influx-induced $[Ca^{2+}]_{ex}$ drop in psoriasis pathology, we established psoriasis model by applying IMQ cream to the ear using *R26-LSL-CEPIAexternal;Itgal-LSL-Clover;Itgb2-LSL-mRuby2;CD4-Cre* mice and injected with SKF96365 or DMSO daily for 7 days (Fig. 9d). Compared with DMSO-treated mice, SKF96365-treated mice exhibited milder ear thickening, less epidermal hyperplasia and decreased number of $\alpha L\beta 2$ positive T cells in skin tissue (Fig. 9e–h), indicating that $Ca^{2+}$ influx-

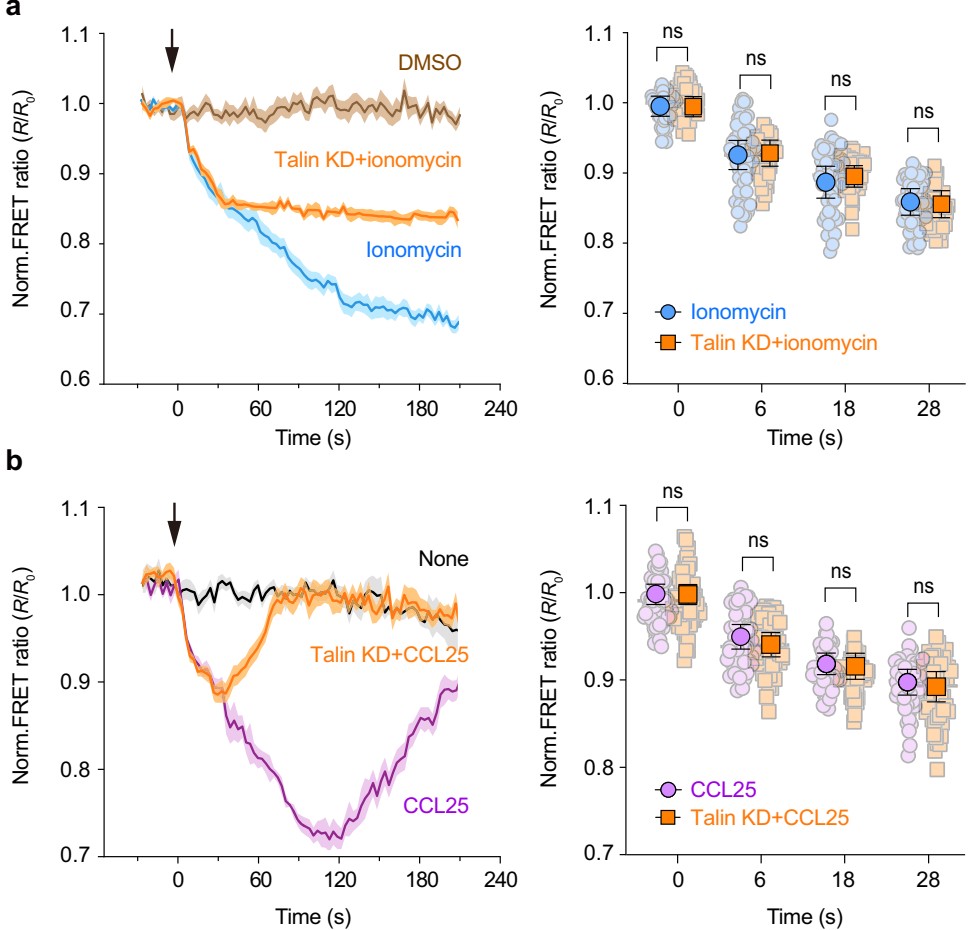

**Fig. 7 | [Ca²⁺]ₑₓ drop-induced integrin αLβ2 activation is independent of integrin inside-out activation signaling.** Splenic T cells were isolated from *R26-LSL-CEPIAexternal;Itgal-LSL-Clover;Itgb2-LSL-mRuby2;CD4-Cre* mice and suspended in buffer containing 1.2 mM Ca²⁺ and 0.6 mM Mg²⁺. 1 μM ionomycin or 0.5 μg/ml CCL25 was added at time point 0. **a** Effect of talin knockdown (Talin KD) on αLβ2 tail FRET ratio change in T cells in response to ionomycin stimulation (left) and the normalized FRET ratios at the representative time points were shown (right) (n = 60 cells from 3 experiments). **b** Effect of talin knockdown (Talin KD) on αLβ2 tail FRET

ratio change in T cells in response to CCL25 stimulation (left) and the normalized FRET ratios at the representative time points were shown (right) (n = 60 cells from 3 experiments). The FRET ratio is normalized to the mean value of cells before the addition of stimuli (R/R₀). The solid lines represent the mean; shaded areas, s.e.m. Data represent the mean ± s.e.m. ns, not significant (unpaired two-tailed Student's *t* test and unpaired two-tailed Welch's unequal variance *t* test in the right panels). Source data are provided as a Source Data file.

---

induced [Ca²⁺]ₑₓ drop and its associated αLβ2 quick activation is critical for T cell entry into psoriatic lesions.

## Discussion

Regulation of integrin affinity for ligands by divalent cations in vitro has been discovered for decades. It has been a mystery whether this kind of regulation can really happen in vivo. Herein, we revealed that Ca²⁺ concentration in T cell membrane-proximal external region can be decreased by chemokine-triggered Ca²⁺ influx to a level to induce αLβ2 activation, providing a mechanism for integrin affinity regulation by Ca²⁺ in vivo. Moreover, chemokine-triggered Ca²⁺ influx induced [Ca²⁺]ₑₓ drop in seconds, which enables immediate activation of integrin when T cells receive chemokine stimulation. Therefore, our findings also provide further mechanisms of integrin rapid activation in a timescale of seconds in vivo.

All living cells have extracellular polysaccharide structures attached. Glycocalyx is the universal term that describes this structure, which is a dynamic surface layer composed of proteoglycans, glycoproteins, and glycosaminoglycans[40]. Glycocalyx is a dense, gel-like meshwork that surrounds the cell and fulfills a multiplicity of functions to cells, including creating a physical and chemical barrier, allowing for buffering extracellular compounds[25]. Especially, the negatively

charged barrier of the glycocalyx impedes the flow of cationic molecules between the cell and its surrounding environment, in particular the divalent cation calcium because of its charge density in the plasma space[41]. Lymphocyte glycocalyx components such as CD45, transmembrane tyrosine phosphatase CD148, transmembrane mucins, and the pericellular matrix generated by hyaluronan and CD44, could shield short receptors through its long and rigid glycocalyx structures[26]. These large glycoproteins on T lymphocytes, presenting a steric barrier perpendicular to the plasma membrane, constrain its lateral diffusion and form a barrier of varying density and length that can curtail access of macromolecules and particulate material to the surface of the cell[42,43]. Several studies have demonstrated that degradation of glycocalyx using endogenous heparinase, hyaluronidase, or neutralizing the negative charge of the glycocalyx by myeloperoxidase can facilitate water and low-density lipoprotein transport, thus enhancing cell permeability[44–46], indicating its crucial role in maintaining the permeable barrier of microcirculation. Indeed, our data showed that Ca²⁺ diffused slowly from solution to T cell surface. Removal of N-glycans on the T cell surface by PNGase F treatment[47] significantly attenuated ionomycin-induced [Ca²⁺]ₑₓ decrease (Fig. 6e, f), suggesting the reduced hinder of Ca²⁺ diffusion from solution to T cell surface after removing cell surface glycocalyx.

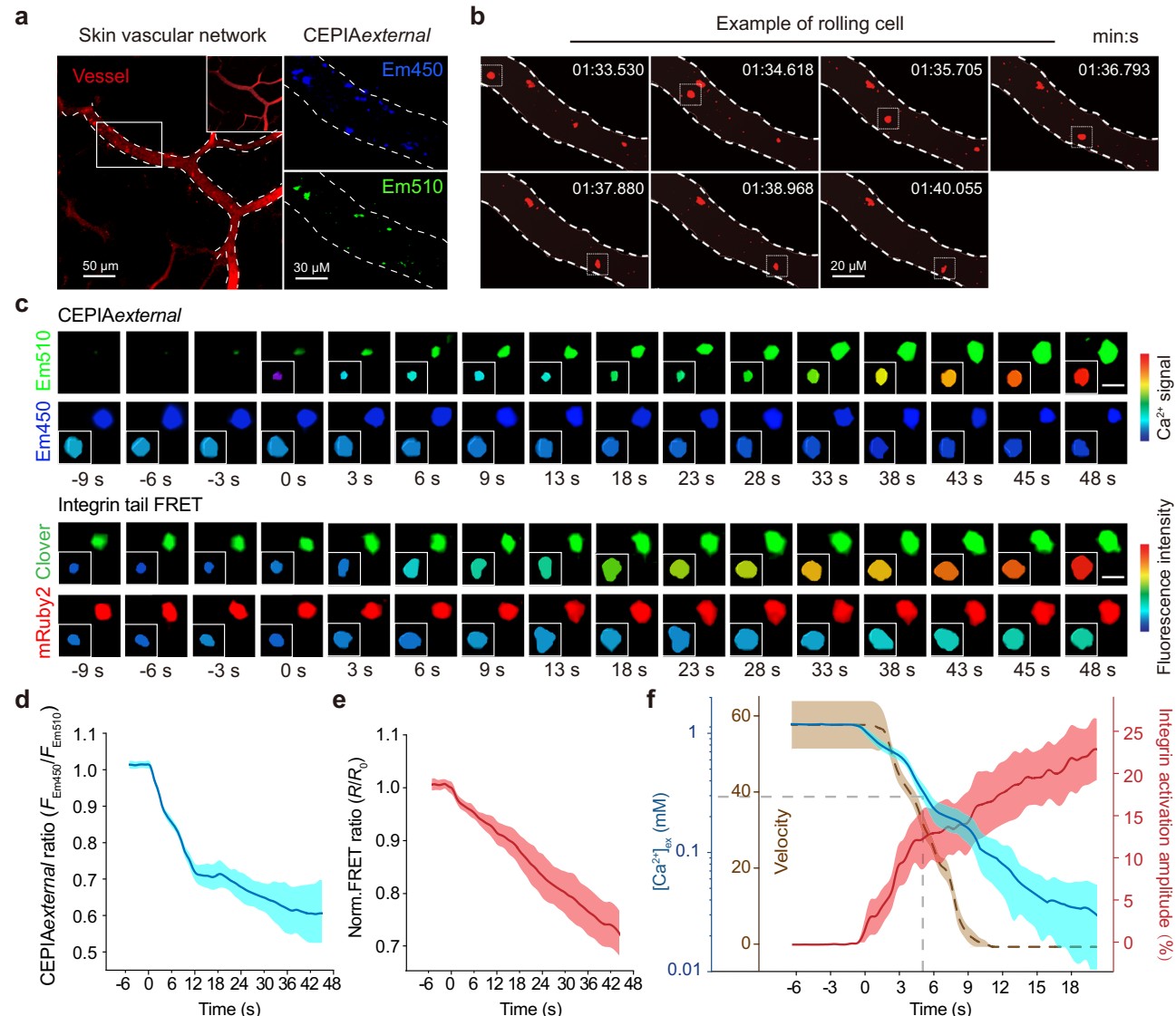

**Fig. 8 | [Ca$^{2+}$]$_{ex}$ dynamics and integrin αLβ2 activation on the surface of homing T cells in mice.** Splenic T cells from *R26-LSL-CEPIAexternal;Itgal-LSL-Clover;Itgb2-LSL-mRuby2;CD4-Cre* mice were labeled with Cell-tracer 647 or Cell-tracer 405 for cell tracking in CEPIA*external* imaging and integrin tail FRET imaging respectively, and then injected via tail vein into recipient mice. **a** Two-photon intravital micrographs of T cells in the psoriasis skin vascular network. Mice were injected with dextran Texas Red and CD31-Alexa Fluor 594 to identify vessels (red). Images are representative of three independent intravital movies. **b** Intravital micrographs of representative rolling T cells labeled with Cell-tracer 647 (red) in vessels of recipient mice. Square highlights rolling cell. Images are from Supplementary Movie 1. Time is shown in min:s. **c** Time series showing the dynamic changes of CEPIA*external* ratio and αLβ2 tail FRET in representative T cells during rolling to arrest transition. Pseudocolor signals were shown as an iso-surface (lower left) pattern based on the original fluorescence (upper right) of T cells. Images are from Supplementary Movie 2. Scale bars, 6 μm. **d, e** Quantification of CEPIA*external* ratio shown in (**d**) and αLβ2 tail FRET ratio dynamic changes of the cells shown in (**e**). The αLβ2 tail FRET ratio is normalized to the mean value of cells in a rolling state ($R/R_0$). Solid lines represent the mean; shaded areas, s.e.m. (*n* = 12). **f** Correlation among [Ca$^{2+}$]$_{ex}$, αLβ2 activation amplitude and rolling velocity of T cells during rolling to arrest transition. Integrin activation amplitude was defined as the extent of a decrease in the normalized FRET ratio compared with the value of rolling T cells. Solid lines represent the mean; shaded areas, s.e.m. (*n* = 12). Source data are provided as a Source Data file.

Divalent cations, such as Ca$^{2+}$ and Mg$^{2+}$, regulate integrin affinity in seconds via binding to metal ion-binding sites in integrin extracellular domains[48]. Integrin αLβ2 has an I domain in the αL subunit, which harbors a metal ion-dependent adhesion site (MIDAS) acting as the primary ligand-binding site[49]. Mg$^{2+}$ in this site forms a direct interaction with a negatively charged residue Glu-34 in ICAM-1[50]. Besides, there is a linear cluster of three metal ion binding sites in the βI domain. MIDAS is located in the center and flanked by two metal ion-binding sites: the adjacent to MIDAS (ADMIDAS) and the ligand-induced metal binding site (LIMBS) also known as synergistic metal ion-binding site (SyMBS)[49]. The divalent cation at βI MIDAS directly coordinates the acidic side chain of Glu-310 in the α7 helix of the αL I domain and

activates the αL I domain by exerting a downward pull on α7 helix[49,51]. LIMBS and ADMIDAS function as positive and negative regulatory sites, respectively[14,49]. The occupancy of ADMIDAS by Ca$^{2+}$ inhibits integrin activation. Removal of Ca$^{2+}$ from ADMIDAS can induce robust integrin activation in seconds. Notably, integrin activation by removal of Ca$^{2+}$ is very rapid because the regulation of integrin affinity by Ca$^{2+}$ does not involve a complicated intracellular signal transduction cascade.

T cells express a variety of calcium channels with selective permeability to Ca$^{2+}$, including voltage-gated calcium channels (e.g., Cav1.1 and Cav2.1) and ligand-gated calcium channels (e.g., store-operated CRAC channels, ORAI1, ORAI2, and ORAI3)[52]. Ca$^{2+}$ is also

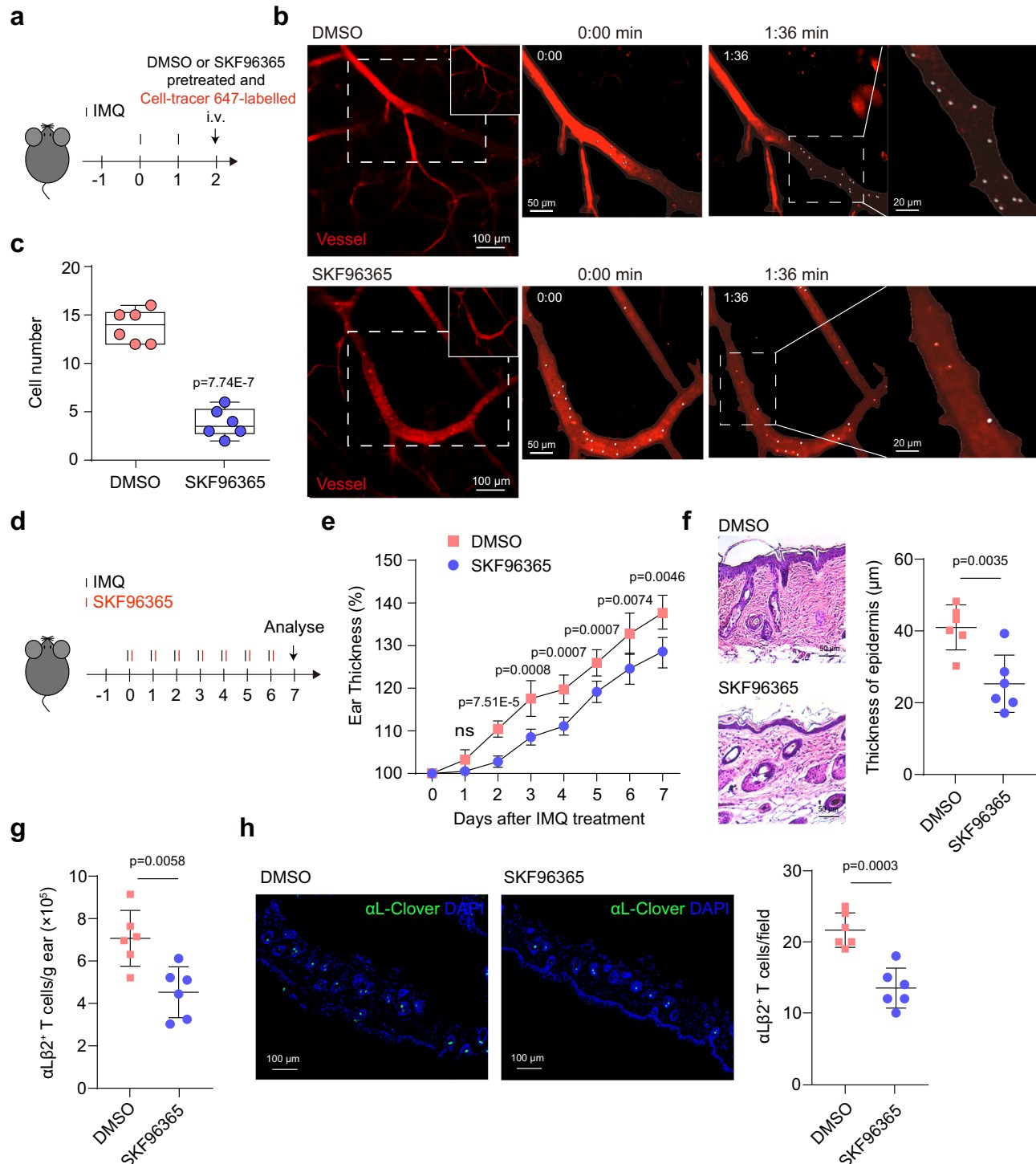

**Fig. 9 | SKF96365 blocks [Ca²⁺]$_{ex}$ drop and alleviates IMQ-induced psoriasis.**
**a–c** 10-week-old mice were treated with 62.5 mg cream containing 5% IMQ onto abdominal skin daily for 2 days (*n* = 6 mice). DMSO and 100 μM SKF96365-pretreated T cells were transferred into mice via tail vein injection. The transferred T cells were labeled with Cell Tracer 647. **a** Schematic diagram of IMQ-induced psoriasis mouse model. **b** Representative intravital micrographs of the transferred T cells arrested in skin postcapillary venules at psoriasiform lesions. Vessels were visualized by dextran Texas Red and CD31-Alexa Fluor 594 (red). Images are from Supplementary Movie 3. Time is shown in min:s. **c** Quantification of the arrested T cells in skin postcapillary venules. Data are presented as box-and-whisker plots showing the median (central line), 25th–75th percentile (bounds of the box), and 5th–95th percentile (whiskers). Each dot represents an individual movie. **d–h** 10-week-old *R26-LSL-CEPIAexternal;Itgal-LSL-Clover;Itgb2-LSL-mRuby2;CD4-Cre* mice

were treated with 18 mg cream containing 5% IMQ onto each ear and injected with 10 mg/kg SKF96365 or DMSO daily for 7 days (*n* = 6 mice). **d** Schematic diagram of IMQ-induced psoriasis and SKF96365 treatment. **e** Changes in ear thickness relative to day 0 of IMQ application. **f** Representative H&E staining of ear sections of DMSO or SKF96365 treated mice on day 7 after IMQ treatment (left). Scale bar: 50 μm. Quantification of epidermal thickness (right). An average of at least 3 measurements per sample was calculated. **g** Quantification of αLβ2⁺ T cells in the ear on day 7 after IMQ treatment. **h** Representative immunofluorescent images of ear skin of mice treated with DMSO or SKF96365 (left). Scale bar: 100 μm. Right: Quantification of αLβ2⁺ T cells in each field (right). An average of at least 3 measurements per sample was calculated. Data represent the mean ± SD. ns, not significant (unpaired two-tailed Student's *t* test). Source data are provided as a Source Data file.

permeable through transient receptor potential channels (TRP channels) on the plasma membrane, which have a relatively non-selective permeability to cations[53]. Upon chemokine stimulation, $Ca^{2+}$ transients are spatially and temporally regulated by the communication between the calcium stores in the ER and membrane distributed calcium channels activated through GPCR signaling[52,54], inducing $Ca^{2+}$ influx achieved through the activation of SOCE, a molecular complex composed of ORAI1 and STIM1[52,55], typically occurring within seconds in a similar time scale as we described in our work[23,31]. It has been reported that different chemokines/cytokines can induce $Ca^{2+}$ influx in a transient or relatively prolonged pattern[56]. For example, CCL19, CCL21, CXCL11 and soluble IL-6Rα can induce an acute and transient increase of cytosolic free $Ca^{2+}$ [57–59], which is comparable to the CCL25-mediated $Ca^{2+}$ influx (Supplementary Fig. 5). By contrast, CCR8 mediates a relatively prolonged extracellular $Ca^{2+}$ entry[56], similar to the effects of CXCL9 and CXCL10 stimulations on $Ca^{2+}$ influx (Supplementary Fig. 8).

In addition to the rapid integrin affinity regulation by chemokine-induced $[Ca^{2+}]_{ex}$ decrease on the T cell surface, previous studies have also reported that certain chemokines can rapidly enhance the avidity and lateral clustering of integrins under shear flow[60,61]. This millisecond-level regulation depends on high concentrations of chemokines and integrin ligands displayed on the vascular endothelium[62]. Both integrin affinity and avidity regulation should contribute to the immediate arrest of leukocytes upon chemokine stimulation.

In summary, our findings reveal a two-phase model for integrin activation by chemokines. The chemokine-triggered $Ca^{2+}$ influx induces $[Ca^{2+}]_{ex}$ drop on the T cell surface, which triggers rapid activation of integrins before the engagement of conventional slow integrin activation via inside-out signaling. $[Ca^{2+}]_{ex}$ drop-induced integrin quick activation fills the gap between initial stimulation and relatively slow inside-out activation of integrin, allowing leukocytes to respond promptly to stimulus and then be arrested at target sites. Blocking $Ca^{2+}$ influx might be a new strategy to regulate leukocyte homing during pathological processes involving aberrant leukocyte trafficking.

## Methods

### Mice
WT C57BL/6 J mice and *CD4-Cre* mice were obtained from Jackson Laboratory. *Itgal-loxP-Stop-loxP-Clover*, *Itgb2-loxP-Stop-loxP-mRuby2,* and *Rosa26-loxP-Stop-loxP-CEPIAexternal* C57BL/6 J mice were generated by Shanghai Biomodel Organism Science & Technology Development Co., Ltd. The genotype of transgenic mice was verified by PCR amplification (Taq DNA Polymerase, Vazyme) and DNA sequencing. Experiments were performed with 7–12 weeks of age mice (both male and female) and littermates were used as controls whenever needed. Equal numbers of male and female animals were employed, and no association of phenotype with sex was detected. All mice were kept in group housing (2–5 mice per cage) in a specific pathogen-free (SPF) facility with controlled environmental conditions of temperature (20–25 °C), humidity (30–70%), and light (a 12- h light/dark cycle) at Center for Excellence in Molecular Cell Science, Chinese Academy of Sciences (CAS). All experiments were conducted under protocols (SIBCB-S323-1802-005) approved by the Institutional Animal Care and Use Committees (IACUCs) of the Center for Excellence in Molecular Cell Science, CAS. Euthanasia was performed using $CO_2$ asphyxiation.

### Flow chamber assay
A polystyrene Petri dish was first incubated with 20 µl ICAM-1–Fc (20 µg/ml) and CCL25 (2 µg/ml) in coating buffer (PBS, 10 mM $NaHCO_3$, pH 9.0) for 1 h at 37 °C, followed by blocking with 2% BSA in coating buffer for 1 h at 37 °C. Cells were then diluted to $1 \times 10^6$ cells/ml in HBSS containing 0.6 mM $Mg^{2+}$ and different concentrations of $Ca^{2+}$ ranging from 0 to 1.2 mM and immediately perfused through the flow chamber at a constant shear stress of 1 dyn/cm² for 1 min. Firmly adherent cells were quantified by counting those that had remained adherent and stationary for at least 10 s.

### Western blotting
T Cells were isolated from *R26-LSL-CEPIAexternal;Itgal-LSL-Clover;Itgb2-LSL-mRuby2;CD4-Cre* mice and lysed with lysis buffer (Cell Signaling Technology #9803) supplemented with protease and phosphatase inhibitor cocktail (Roche #04693159001, #04906837001) on ice. Immunoblot analysis was then conducted with antibodies against integrin β1 (1:2000, Abcam #Ab52971), HA tag (1:1000, Cell Signaling Technology #3724 S), β-actin (1:5000, ABclonal #AC004), Talin (1:500, Sigma-Aldrich #T3287). Secondary antibodies used were Goat Anti-Mouse IgG (H + L) HRP (1:5000, Multi Sciences #GAM0072) and Goat Anti-Rabbit IgG (H + L) HRP (1:5000, Multi Sciences #GAR0072). The uncropped scans of the blots are provided within the Source Data.

### Integrin tail FRET imaging
T cells from *R26-LSL-CEPIAexternal;Itgal-LSL-Clover;Itgb2-LSL-mRuby2;CD4-Cre* mice were washed three times with 5 mM EDTA, and then washed twice with KCl/MOPS buffer (130 mM KCl, 50 mM MOPS, pH 7.2) to remove EDTA. Cells were subsequently resuspended in KCl/MOPS buffer containing 1.2 mM $Ca^{2+}$ and 0.6 mM $Mg^{2+}$. 5 mM EGTA (HY-D0861, MCE) was added at time point 0 to chelate $Ca^{2+}$ in the solution. For BAPTA-AM (A1076, Sigma) or SKF96365 (HY-100001, MCE) treatment, cells were pretreated with 100 µM BAPTA-AM or 100 µM SKF96365 for 30 min at 37 °C. The cells were then seeded on poly-L-lysine (100 µg/ml) substrates in serum-free medium for 3 min at room temperature, then followed by stimulation with 1 µM ionomycin (Cell Signaling Technology #9995 S), 0.5 µg/ml CCL25 or KCl/MOPS buffer control. Time-lapse images at a rate of one frame per 3 s were captured using an inverted Nikon A1 confocal microscope equipped with a 60 × oil objective and the following filters (Ex = excitation, Em = emission): Ex 485/30 nm, Em 530/40 nm for FRET donor-Clover, and Ex 485/30 nm, Em 595/70 nm for FRET acceptor-mRuby2. The raw data was collected and analyzed using NIS-Elements software (Nikon), and then regions of interest (ROIs) were drawn along the plasma membrane to form a round circle for reporter responses and a cell-free region for background measurements. Emission ratios were obtained by calculating background-subtracted FRET intensities divided by background-subtracted donor intensities ($F_{mRuby2}/F_{Clover}$). Time-course ratio measurements were normalized to baseline pre-stimulation values ($R/R_0$). Graphs were plotted using GraphPad Prism 9 (GraphPad Software).

### $Ca^{2+}$ concentration measurement using CEPIA imaging
To monitor membrane-proximal external $Ca^{2+}$ concentration ($[Ca^{2+}]_{ex}$), we established a plasma membrane-anchored external $Ca^{2+}$ biosensor named CEPIA*external*, which was modified from GEM-CEPIA1*er*[22]. Firstly, we deleted the ER-targeting sequence of the original GEM-CEPIA1*er*. Then, we added the PDGFR transmembrane (PDGFR-TM) sequence to the C-terminus of CEPIA to ensure the extracellular location of the modified biosensor. For $[Ca^{2+}]_{ex}$ measurement, T cells from *R26-LSL-CEPIAexternal;Itgal-LSL-Clover;Itgb2-LSL-mRuby2;CD4-Cre* mice were prepared the same as in the FRET imaging experiment. Time-lapse live cell images were captured at a rate of one frame per 3 s using an inverted Nikon A1 confocal microscopy equipped with a 60 × oil objective. The Ex/Em filter settings were 377 ± 25 nm/466 ± 20 nm and 377 ± 25 nm/520 ± 17.5 nm. To evaluate the changes in $[Ca^{2+}]_{ex}$ at the plasma membrane, a circular ROI was delineated along the external submembrane regions.

For ER $Ca^{2+}$ concentration ($[Ca^{2+}]_{er}$) measurement, T cells from WT mice transfected with lentivirus encoding CEPIA*er*[22] were used for confocal microscopy. ER was delineated by a ROI.

To monitor membrane-proximal internal $Ca^{2+}$ concentration ($[Ca^{2+}]_{in}$), we established a plasma membrane-anchored internal $Ca^{2+}$

biosensor by adding PDGFR-TM sequence to the N-terminus of CEPIA (CEPIA*internal*). For $[Ca^{2+}]_{in}$ measurement, T cells from WT mice transfected with lentivirus encoding CEPIA*internal* were used for confocal microscopy. To evaluate the $Ca^{2+}$ changes at the intracellular membrane-proximal locations, a circular ROI was delineated along the internal submembrane regions.

ImageJ and Nikon NIS-Elements were employed to quantify the fluorescence intensity of the ROIs, and the CEPIA ratio ($F_{Em450}/F_{Em510}$) was calculated. Graphs were plotted using GraphPad Prism 9 (Graph-Pad Software). The obtained $[Ca^{2+}]$-CEPIA ratio relationship was fitted by the following Hill equation using a least-square method in OriginPro 8 software.

$$R = R_{min} + (R_{max} - R_{min}) * [Ca^{2+}]_{ex}^n / (K^n + x^n)$$

$R = F_{Em450}/F_{Em510}$, $n = 1.33$ and $K = 0.43$. $R_{min} = 0.624$ and $R_{max} = 1.13$ for T cells in KCl/MOPS buffer. $[Ca^{2+}]$ was calculated with the CEPIA ratio by the equation.

For cytosolic $Ca^{2+}$ imaging using Fluo-4, cells were loaded with 5 μM Fluo-4 AM (F14201, Invitrogen) in 0.1% BSA-supplemented KCl/MOPS buffer for 40–60 mins at room temperature, followed by replacing the loading solution with KCl/MOPS buffer without BSA before imaging.

## Extracellular $Ca^{2+}$ measurement using Rhod Red probe

T cells from WT mice were prepared the same as in the FRET imaging experiment to remove any divalent metal ions, then were subsequently resuspended in KCl/MOPS buffer containing 1.2 mM $Ca^{2+}$ and 0.6 mM $Mg^{2+}$. The dynamic changes of extracellular $Ca^{2+}$ around T cells were quantified using an Amplite™ Fluorimetric Calcium Quantitation Kit (36360, AAT Bioquest). The cell suspension was mixed with the Rhod Red stock solution (200×) and incubated in the dark for 10 min for color development, followed by stimulation with ionomycin or CCL25 and time-lapse imaging using a Nikon A1 confocal microscope. The Ex/Em filter settings were 540 nm and 590 nm, with 570 nm detected as background at time 0 s. To evaluate the changes in $[Ca^{2+}]_{ex}$ at the plasma membrane, a circular ROI was delineated along the external submembrane regions. For the analysis of the fluorescence imaging, rainbow colors were used to represent the 590 nm signal changes.

For PNGase F treatment, T cells were incubated with 1 ml serum-free medium containing 1000 units of PNGase F (P2318S, Beyotime) for 24 h at 37 °C to release all types of N-glycans before extracellular $Ca^{2+}$ measurement.

## Confocal imaging of CEPIA-labeled agarose beads

To compare the $Ca^{2+}$ diffusion rates from solution to the surfaces of agarose beads and T cells, CEPIA-labeled agarose beads were generated by incubation of anti-HA agarose beads (KTSM1305, Alpalifebio) with the 293 T culture medium containing soluble CEPIA proteins at 4 °C for 4 h, followed by three washes with 5 mM EDTA and two washes with KCl/MOPS buffer (130 mM KCl, 50 mM MOPS, pH 7.2) to eliminate EDTA. CEPIA-labeled agarose beads and CEPIA*external* expressing T cells were positioned in 35 mm glass-bottom dishes, time-lapse images were acquired using Nikon A1 confocal microscope.

## High sensitivity structured illumination microscope (HiS-SIM) imaging

HiS-SIM (Guangzhou Computational Super-resolution Biotech) was used to acquire images to show the CEPIA*external* and CEPIA*internal* distribution in T cells. The mouse T cells expressing either CEPIA*external* or CEPIA*internal* were stained with 5 μg/mL plasma membrane dye FM 4-64FX (F34653, ThermoFisher) and then imaged using HiS-SIM with a ×100/1.5 NA oil immersion objective (Olympus). The raw image was exposed for 10 ms and captured with a resolution of 256 × 256 pixels. HIS-SIM is controlled by its own software Imager (v1.1.23d).

## Silencing of talin in T cells

Silencing of mouse talin in T cells was achieved by shRNA. Recombinant lentiviruses expressing scramble shRNA (5′-CCTAAGGT-TAAGTCGCCCTCG-3′) or shRNAs that annealed to talin (shTalin-1#: 5′-GCAGAAGGGAGAGCGTAAGAT-3′; shTalin-2#: 5′-GAAGCACAGAGC CGATTGAAT-3′) were used. The silencing of talin was confirmed by immunoblotting 48 h post-transfection.

## Psoriasis mouse model

C57BL/6 mice 8–10 weeks old were topically administered a daily dose of 62.5 mg of IMQ cream (5%) to the depilated abdominal area for two consecutive days. The mice were then used for two-photon intravital imaging.

## Two-photon intravital imaging

$2.5 \times 10^7$ T cells from *R26-LSL-CEPIAexternal;Itgal-LSL-Clover;Itgb2-LSL-mRuby2;CD4-Cre* mice were labeled with Cell-tracer 647 (C34572, Invitrogen) for CEPIA*external* imaging or with Cell-tracer 405 (C34568, Invitrogen) for integrin tail FRET imaging, and then injected via tail vein (i.v.) into IMQ-induced psoriasis recipient mice. For SKF96365 treatment, cells were pretreated with or without 100 μM SKF96365 for 30 min at 37 °C and then injected into the recipient mice. To visualize vessels, Alexa Fluor 594-conjugated anti-mouse CD31 and dextran Texas Red (D3328, Invitrogen) were injected i.v. immediately before the surgery. Then mice were anesthetized with tribromoethanol (250–500 mg/kg) intraperitoneally (i.p.). After the removal of abdominal fur, a 4–5 cm skin incision was made at the center abdomen, then the subcutaneous venules were exposed after gently peeling and skin turning over, followed by attaching the skin to a plastic coverslip and immersing it in warm saline. Imaging was conducted at room temperature using two-photon microscopy (FVMPE-RS, OLYMPUS) equipped with two infrared lasers (MAITAI HPDS-OL: 690 nm-1040 nm; INSIGHT X3-OL: 690 nm-1300 nm). The MAITAI laser was adjusted to 800 nm for excitation of BV450 or BV510. INSIGHT laser excitation was tuned to 1100 nm for simultaneous excitation of FITC, APC, or PE. A 25 × 1.05 NA water lens (XLPlan N, OLYMPUS) coupled to a 4-color detector array was utilized to detect emitted light.

For movie acquisition, images were collected at a rate of 1 frame per second (FPS) with 512 × 512 pixels or 20 FPS with 256 × 256 pixels. Six random field-of-views were collected per animal within the subcutaneous venules. Movies were processed and analyzed by IMARIS 9.5 (Bitplane). The vessel surface was extracted using IMARIS surface built-in function based on dextran Texas Red and CD31-Alexa Fluor 594 signals. To visualize single cells, T cells from donor mice were enriched with negative magnetic selection (EasySepTM) and labeled with Cell-tracer 647 or Cell-tracer 405. Spot points were then created and tracked over time with the IMARIS spot built-in function.

## Statistics and reproducibility

All data were tested using the Shapiro-Wilk and Kolmogorov-Smirnov normality tests. For Gaussian data, pairwise comparisons were performed using unpaired Student's *t* test or Welch's unequal variance *t* test after variance homogeneity tests using the F test. Comparisons between three or more groups were performed using ordinary one-way ANOVA or Brown-Forsythe and Welch one-way ANOVA, followed by Dunnett's test for multiple comparisons after variance homogeneity tests using Bartlett's test and Brown-Forsythe test. All statistical analysis was calculated using GraphPad Prism 9 (GraphPad Software). Differences with a *p*-value below 0.05 were considered statistically significant. Unless otherwise noted, experiments were repeated at least three times with similar results. Average time courses and

micrographs shown in the figures depict individual representative experiments.

**Reporting summary**

Further information on research design is available in the Nature Portfolio Reporting Summary linked to this article.

## Data availability

All data generated in this study are available in the paper in the Source Data file or from the corresponding author on request. Source data are provided with this paper.

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

## Acknowledgements

This work was supported by grants from the National Key R&D Program of China (2020YFA0509102, 2020YFA0509000 to J.F.C.), National Natural Science Foundation of China (31830112, 32030024 to J.F.C., 92369102, 32170769 to C.D.L., 82171804 to Y.H.Z.), Program of Shanghai Academic Research Leader (19XD1404200 to J.F.C.), China Postdoctoral Innovative Talent Support Program (BX20190345 to S.H.W.), China Postdoctoral Science Foundation (2020M671262 to S.H.W.), Shanghai Rising-Star Program (21QA1409700 to C.D.L.), the Science and Technology Innovation Action Plan of Natural Science Foundation of Shanghai (23ZR1466200 to C.D.L.), National clinical key specialty construction project of China (Z155080000004 to C.D.L.), Shanghai Research Center of Rehabilitation Medicine (Top Priority Research Center of Shanghai) (2023ZZ02027 to C.D.L.), Shanghai Disabled Persons' Federation Key Laboratory of Intelligent Rehabilitation Assistive Appliance and Technology, the Fundamental Research Funds for the Central Universities (22120240034, 22120240435 to C.D.L.), and Peak Disciplines (Type IV) of Institutions of Higher Learning in Shanghai. The authors gratefully acknowledge the support of the SA-SIBS scholarship program and Eastern Talents Program-Youth Project.

## Author contributions

Y.L., S.H.W., Y.H.Z., C.D.L., and J.F.C. designed experiments. Y.L., S.H.W., C.D.L., Y.H.Z., Z.Y.L., Y.Z.Z., K.Z., X.Y.L., M.W.H., X.C.P., S.Y.C., Y.J.Z., and M.Y.Y. performed experiments and analyzed data. Y.L., S.H.W., C.D.L., Y.H.Z., G.X.G., Y.A. Z., and J.F.C. interpreted results. The manuscript was drafted by Y.L., S.H.W., C.D.L., and edited by J.F.C.

## Competing interests

The authors declare no competing interests.
