## [Peer Review File · Nature Communications]

Ca²⁺ transients on T cell surface trigger integrin rapid activation on a second timescaleEditorial Note: Figures have been redacted from this peer review file.

REVIEWER COMMENTS

Reviewer #1 (Remarks to the Author):

In this study Li et al. address a gap that has been left open in the study of kinetics of integrin activation, explaining how a very rapid activation of $\alpha\text{L}\beta\text{2}$, within seconds, by a drop in external calcium concentration leads to the immediate arrest of T cells at the site of high CCL25 chemokine concentration, therefore earlier to the slow inside-out activation of integrins, which takes minutes to develop.

To do this, they develop a FRET probe that reports on $\alpha\text{L}\beta\text{2}$ activation and another probe (CEPIA) that measures local calcium concentration, and try to establish a correlation between the two. These constructs are engineered by CRISPR and mouse genetics to express them together in mouse T cells.

The strategy appears appropriate and robust and the overall results also seem consistent that an initial drop in surface calcium induces the initial activation of $\alpha\text{L}\beta\text{2}$ by ionomycin or CCL25, which is unaltered by the use of calcium chelators or calcium influx inhibitors. However, the conclusions of most experiments rely on the effectiveness of the calcium probe (CEPIA) and it is important that this is validated in the system used (R26-LSL-CEPIA;Itgal-LSL-Clover;Itgb2-LSL-mRuby2;CD4-Cre mouse T cells).

Major points

1. I was surprised to see that the concentration of calcium remains low in the outer surface of the membrane for a long time after ionomycin administration even though it stabilizes inside the cell after 3 min (Fig. 3A-D). Calcium influx stops within 10 sec of adding ionomycin. The results in Fig. 4 using CCL25 confirm that the drop in concentration of surface calcium is linked to calcium influx and that there is a recovery in surface calcium after 30 sec (Fig. 4B-C). How can you explain that with the maintenance of calcium in

solution for the whole of the experiments (Fig. 3), calcium values at the surface of the plasma membrane remain low?

2. The original CEPIA biosensor mentioned had ER-targeting sequences (ref. 22); it is unclear what modifications, other than the insertion of the PDGFR transmembrane domain, were made to target this complex to the outer surface of the plasma membrane. Authors must detail the construction of the calcium sensor and present evidence that it is located on the extracellular side of the plasma membrane.

Minor points

3. In Fig. 3H, the statistics should not directly compare the ionomycin bars (control vs BAPTA) as appears to have been done, but rather the normalization of these columns to their respective DMSO control. For consistency, same with Fig. 4H.

4. Lines 206-207: "Next, we investigated the role of $[Ca^{2+}]_{ex}$ decrease-induced $\alpha L\beta 2$ quick activation in T cell homing...". Although it is presumed, it cannot be stated that integrin activation is due to $[Ca^{2+}]_{ex}$ decrease, as this is a variable that is being evaluated and needs validation. This sentence needs to be rephrased. In subsequent similar sentences, namely in relation to Fig. 7, keep in mind that the $[Ca^{2+}]_{ex}$ decrease is presumed.

5. The sentences in lines 250-252 and 269-271 are poorly constructed, contain no conclusions or are not terminated (alternatively remove "Because" in both cases).

Reviewer #2 (Remarks to the Author):

The manuscript by Li and colleagues investigates mechanisms by which lymphocytes are arrested at target sites along blood vessel linings and finally transmigrate through the vessel wall. Typically, this is achieved via a chemokine and integrin-mediated mechanism. In this study the authors developed assays to study integrin activation via FRET and extracellular calcium via localization of biosensors to the extracellular side of the lymphocyte plasma membrane. Based on their experimental data they put forward that chemokine or ionomycin stimulation of lymphocytes leads to decreases in extracellular calcium close to

the plasma membrane that underlies integrin activation and T-cell arrest.

My main criticism is directed to the calcium imaging data. It is implausible that decreases in extracellular calcium close to the plasma membrane could be achieved and maintained in a high calcium containing environment for the indicated periods of time (many seconds). How is this supposed to work? Could there be any extracellular matrix components that insulate the calcium space close to the membrane from the overall volume? The authors do not comment or discuss this at all. Without any further sound evidence that this could actually be true I do not believe it at all. My impression is that the authors observe some kind of artefact of the membrane-anchored indicator.

Reviewer #3 (Remarks to the Author):

Li and colleagues have studied the Ca²⁺-dependent mechanism causing immediate rolling arrest in lymphocytes exposed to chemokines. By using an array of molecular and biophysical methods to probe [Ca²⁺]_{ex} and manipulate intracellular signals, the authors conclude that the rapid chemokine-dependent lymphocyte arrest mainly depends on a [Ca²⁺]_{ex} drop occurring within seconds, which activates αLβ2 integrin and thus cell adhesion. The paper is generally interesting and the authors' suggestions original. However, to be fully convincing, the conclusion that the quick effect on cell arrest is essentially due to [Ca²⁺]_{ex} drop would require some other lines of evidence.

1) My main concern is that proving the above conclusion would need more precise measurement of internal submembrane calcium, and/or more specific test of its effects. In general, it is difficult to completely block local submembrane intracellular calcium elevations merely by calcium chelators (and it is not clear from the Results if the authors have specific evidence about this in the present system). In addition, the degree of proximity between the inner mouth of the calcium pore and the inner domains of the other membrane proteins implicated in the studied effect is unknown, to the best of my knowledge. Thus (lines 193-194), SKF9665 may be more effective because, by blocking calcium influx, cuts at the root not only the [Ca²⁺]_{ex} drop but also possible local submembrane effects of calcium.

To fully demonstrate that the effect is essentially caused by external calcium depletion, one would have to apply some of the following approaches: a) attempting to measure local

internal calcium changes as the authors did for the external; b) block/inactivate/delete the possible submembrane targets of rapid calcium effects (are these targets known?); d) have the authors ever attempted to release caged calcium intracellularly and test the kinetics of leukocyte arrest? e) would it be possible to carry out complementary experiments, in the presence of increased $[Ca^{2+}]_{ex}$?

2) Based on the results with SKF9665, do the authors think that the quick chemokine-induced calcium influx essentially depends on store-operated calcium entry (SOCE) activation? This is not very clear in the text. If so, is the known kinetics of SOCE activation (TRP, Ora-1?) fully consistent with the quick time course of extracellular calcium depletion?

3) Fig. 1 and 2. Does nominal 0 Ca^{2+} really correspond to $[Ca^{2+}]_o = 0$? Calcium contaminations are known to be present in salts and water (in the order of 0.1 mM). Did the authors ever test EGTA as in panel F? The ratio observed in the presence of EGTA (approximately 0.85) does not correspond to the ratio observed in 0 calcium in panel E. Could the flat part of the CEPIA ratio below 0.1 mM partly artefactual? Perhaps a control with EGTA would be also useful here.

Discussion. The first part of the paper is generally clearly written, but the writing style deteriorates considerably in the Discussion. In general, this section seems somewhat perfunctory.

Minor issues.

Introduction/p. 5/elsewhere. It should be perhaps made clear at the outset that the authors are referring to free calcium and magnesium, not total blood concentration. In fact, the titles of the cited references (16-17) mention 'serum' not blood.

Line 88. ICAM-1 is wrongly defined as 'intracellular adhesion molecule-1' (instead of 'intercellular....etc.), which could make the paragraph puzzling to the general reader. Moreover, the experimental system should be briefly explained (ICAM-1 was immobilized

on culture dishes), as the Fig. 1 legend is also not very clear.

Line 139. 'western' or 'Western'?

Line 171. 'BAMPTA'

p. 9 Lines 180 (...was not affected..) and 183 (...partially prevented...) seem somewhat contradictory. The entire paragraph could be reworded for clarity.

Line 187. The action and specificity of SKF96365 should be explained here.

Line 325. Because Dyn/cm² are the physical dimensions of pressure and not flow rate, this should be perhaps briefly explained.

Statistics. In Fig. 1, it is not clear which pairs were compared with the post-hoc test.

NCOMMS-23-42688A

Response to reviewers' comments

We are grateful to the comments by all reviewers for the novelty and general interests of the paper. The reviewers made some very constructive comments that are very helpful to revising the manuscript. We have addressed all the raised concerns with the suggested experiments or better interpretation of our data, which should help to further improve the overall quality of the manuscript. All of the revisions have been highlighted in red in the revised manuscript. The point-by-point response is shown as below.

Reviewer #1: In this study Li et al. address a gap that has been left open in the study of kinetics of integrin activation, explaining how a very rapid activation of $\alpha L\beta 2$, within seconds, by a drop in external calcium concentration leads to the immediate arrest of T cells at the site of high CCL25 chemokine concentration, therefore earlier to the slow inside-out activation of integrins, which takes minutes to develop. To do this, they develop a FRET probe that reports on $\alpha L\beta 2$ activation and another probe (CEPIA) that measures local calcium concentration, and try to establish a correlation between the two. These constructs are engineered by CRISPR and mouse genetics to express them together in mouse T cells.

The strategy appears appropriate and robust and the overall results also seem consistent that an initial drop in surface calcium induces the initial activation of $\alpha L\beta 2$ by ionomycin or CCL25, which is unaltered by the use of calcium chelators or calcium influx inhibitors. However, the conclusions of most experiments rely on the effectiveness of the calcium probe (CEPIA) and it is important that this is validated in the system used (R26-LSL-CEPIA;Itgal-LSL-Clover;Itgb2-LSL-mRuby2;CD4-Cre mouse T cells).

Point 1. I was surprised to see that the concentration of calcium remains low in the outer surface of the membrane for a long time after ionomycin administration even though it

stabilizes inside the cell after 3 min (Fig. 3A-D). Calcium influx stops within 10 sec of adding ionomycin. The results in Fig. 4 using CCL25 confirm that the drop in concentration of surface calcium is linked to calcium influx and that there is a recovery in surface calcium after 30 sec (Fig. 4B-C). How can you explain that with the maintenance of calcium in solution for the whole of the experiments (Fig. 3), calcium values at the surface of the plasma membrane remain low?

Response: We thank the reviewer for this very constructive comment. The confusion is mainly caused by the conflict of the two results: 1) The concentration of Ca^{2+} in the outer surface of the plasma membrane ($[\text{Ca}^{2+}]_{\text{ex}}$) remained low for at least 180 seconds after ionomycin administration (Fig. 3a-c), which suggests a continuous Ca^{2+} influx; 2) The cytosolic Ca^{2+} concentration ($[\text{Ca}^{2+}]_{\text{cyto}}$) only showed a short-time increase then remained stable (Supplementary Fig. 8a). If ionomycin induces a continuous Ca^{2+} influx, the inflow Ca^{2+} should be stored somewhere in the cells. Given the endoplasmic reticulum (ER)'s role as the main calcium reservoir¹, we hypothesize that ER stores the inflow Ca^{2+} therefore maintains a stable level of $[\text{Ca}^{2+}]_{\text{cyto}}$. To validate our hypothesis, we have expressed ER-targeting CEPIA (CEPIA_{er})² in T cells and employed the CEPIA_{er} imaging to monitor the level of Ca^{2+} in ER ($[\text{Ca}^{2+}]_{\text{er}}$) post ionomycin administration (Supplementary Fig. 8b,c). The results showed that $[\text{Ca}^{2+}]_{\text{er}}$ increased continuously upon ionomycin addition (Supplementary Fig. 8c). Thus, ionomycin induces a continuous Ca^{2+} influx for at least 180 seconds, which induces a short time increase in $[\text{Ca}^{2+}]_{\text{cyto}}$ level and continuous increase in $[\text{Ca}^{2+}]_{\text{er}}$.

[figure redacted]

Supplementary Fig. 8: Effects of ionomycin treatment on intracellular Ca^{2+} dynamics in T cells.

Splenic T cells from WT mice transfected with CEPIA_{er} were suspended in buffer containing 1.2 mM Ca^{2+} and 0.6 mM Mg^{2+} . Ionomycin (final concentration 1 μM) were added at time point 0.

a, Cytosolic Ca^{2+} ($[\text{Ca}^{2+}]_{\text{cyto}}$) change was detected by Fluo-4 in T cells pretreated with 100 μM BAPTA-AM or DMSO vehicle control in response to stimulation with 1 μM ionomycin.

b, Representative pseudocolour images of CEPIA_{er} ratio in response to stimulation with 1 μM ionomycin. The first two images show the distribution of CEPIA_{er} in Em450 and Em510 channels. Scale bar, 3 μm .

c, Time course of CEPIA_{er} ratio change (left panel) and the corresponding change of Ca²⁺ in ER ([Ca²⁺]_{er}) (right panel) in response to stimulation with 1 μM ionomycin. The solid lines represent the mean; shaded areas, s.e.m in **a** and **c**. *n* = 12 cells from 3 experiments.

In addition, the slow diffusion of Ca²⁺ from solution to the surface of T cells might also contribute to the Ca²⁺ influx-induced low level of [Ca²⁺]_{ex}. It is known that most cells including lymphocytes are covered by a dense glycocalyx, which resides extracellularly on the cell membrane, surrounding the cell like a cloak^{3, 4}. This special structure contributes to the formation of a physical and charged barrier modulating the flow of low molecular substrates or ions into and out of the cells^{5, 6, 7}, which may result in the delay of Ca²⁺ diffusion from the solution to the external surface of T cells. We have performed a new experiment to compare the diffusion rate of Ca²⁺ from solution to the surfaces of CEPIA-labeled agarose beads and T cells. The results clearly showed that addition of 1.2 mM Ca²⁺ into the solution rapidly increased [Ca²⁺]_{ex} on the surface of agarose beads (Supplementary Fig. 7). By contrast, [Ca²⁺]_{ex} on T cells surface showed slowly increase. These data indicate that Ca²⁺ diffuse slowly from solution to T cells surface.

[figure redacted]

Supplementary Fig. 7: Ca²⁺ in the solution diffuses more quickly to the surface of agarose beads than to cell surface.

a, Immunoblot analysis of soluble CEPIA-HA protein with anti-HA tag antibody in culture media of 293T cells transiently transfected with CEPIA-HA or vector control.

b-c, Splenic T cells isolated from *R26-LSL-CEPIAexternal;Itgal-LSL-Clover;Itgb2-LSL-mRuby2;CD4-Cre* mice or CEPIA-HA-labeled anti-HA agarose beads were suspended in buffer containing 0 mM Ca²⁺ and 0.6 mM Mg²⁺. 1.2 mM Ca²⁺ (final concentration) were added at time point 0.

b, Representative pseudocolour images of CEPIA_{external} ratio on the surface of T cells or CEPIA-HA ratio on the surface of CEPIA-HA-labeled agarose beads in response to addition of Ca²⁺. The first two images show the distribution of CEPIA_{external} and CEPIA-HA in Em450 and Em510 channels. Scale bar, 3 μm.

c, Time course of CEPIA_{external} ratio change and the corresponding change of [Ca²⁺]_{ex} on T cell surface or changes of CEPIA-HA ratio and [Ca²⁺]_{ex} on CEPIA-HA-labeled agarose bead surface in response to addition of Ca²⁺.

The solid lines represent the mean; shaded areas, s.e.m. in **c**. *n* = 12 cells and *n* = 12 beads from 3 experiments.

Taken together, the continuous Ca^{2+} influx and slow Ca^{2+} diffusion from solution to T cell surface results in the a persistent low $[\text{Ca}^{2+}]_{\text{ex}}$ on T cells surface. The new results have been included in the revised manuscript (please refer to lines 280-304 on pages 12-13).

Point 2. The original CEPIA biosensor mentioned had ER-targeting sequences (ref. 22); it is unclear what modifications, other than the insertion of the PDGFR transmembrane domain, were made to target this complex to the outer surface of the plasma membrane. Authors must detail the construction of the calcium sensor and present evidence that it is located on the extracellular side of the plasma membrane.

Response: We felt sorry that we did not describe the plasma membrane-anchored external CEPIA (*CEPIAexternal*) clearly. In this study, we deleted the ER-targeting sequence of the original CEPIA biosensor (*GEM-CEPIA1er*)² and added PDGFR transmembrane (PDGFR-TM) sequence to the C-terminus of CEPIA. We have described the details for the construction of *CEPIAexternal* in the revised manuscript (lines 129-130 on page 6 and lines 426-431 on page 19). In addition, we have also used super-resolution microscopy to show that *CEPIAexternal* biosensors are located on the extracellular side of plasma membrane (Fig. 2d). The new results have been included in the revised manuscript (lines 143-145 on page 7).

Point 3. In Fig. 3H, the statistics should not directly compare the ionomycin bars (control vs BAPTA) as appears to have been done, but rather the normalization of these columns to their respective DMSO control. For consistency, same with Fig. 4H.

Response: Thanks for the comment. We felt sorry for the confusion. In Fig. 3h, the cells were pretreated with BAPTA-AM or DMSO, then followed by stimulation with ionomycin. We want to show the following points with these data. 1) BAPTA-AM treatment did not affect the basal cell adhesion mediated by integrin $\alpha\text{L}\beta 2$; 2) ionomycin treatment promoted

cell adhesion; 3) BAPTA-AM treatment partially inhibited the ionomycin-enhanced cell adhesion. Therefore, we have replotted the Fig. 3h to make these points clear. Similar revision was applied to Fig. 4h. The new Fig. 3h and 4h have been updated in the revised manuscript.

[figure redacted]

Fig. 3h: Effect of ionomycin treatment on the adhesion of T cells pretreated with 100 μ M BAPTA-AM or DMSO vehicle control to the immobilized ICAM-1 (20 μ g/ml) substrates at a wall shear stress of 1 dyn/cm² ($n = 3$).

Fig. 4h: Adhesion of T cells pretreated with 100 μ M BAPTA-AM, 100 μ M SKF96365 or vehicle control to the immobilized ICAM-1 (20 μ g/ml) alone or ICAM-1 (20 μ g/ml) plus CCL25 (2 μ g/ml) substrates at a wall shear stress of 1 dyn/cm² ($n = 3$).

Data represent the mean \pm s.e.m. * $P < 0.05$; ** $P < 0.01$; *** $P < 0.001$; **** $P < 0.0001$; ns, not significant (one-way ANOVA with Dunnett's test).

Point 4. Lines 206-207: “Next, we investigated the role of $[Ca^{2+}]_{ex}$ decrease-induced $\alpha L\beta 2$ quick activation in T cell homing...”. Although it is presumed, it cannot be stated that integrin activation is due to $[Ca^{2+}]_{ex}$ decrease, as this is a variable that is being evaluated and needs validation. This sentence needs to be rephrased. In subsequent similar sentences, namely in relation to Fig. 7, keep in mind that the $[Ca^{2+}]_{ex}$ decrease is presumed.

Response: Thanks for the comments. As suggested, we have rephrased all related sentences in the revised manuscript by changing “ $[Ca^{2+}]_{ex}$ decrease-induced $\alpha L\beta 2$ quick activation” to either “ Ca^{2+} influx-induced $[Ca^{2+}]_{ex}$ drop and its associated $\alpha L\beta 2$ quick activation” or “ Ca^{2+} influx-induced $[Ca^{2+}]_{ex}$ drop” (lines 214-217, 233 and 236-238 on page 10, lines 246 and 252 on page 11).

Point 5. The sentences in lines 250-252 and 269-271 are poorly constructed, contain no conclusions or are not terminated (alternatively remove “Because” in both cases).

Response: Thanks for the comments. We have rephrased all these sentences in the revised manuscript as below.

1) Moreover, chemokine-triggered Ca^{2+} influx induced $[\text{Ca}^{2+}]_{\text{ex}}$ drop in seconds, which enables immediate activation of integrin when T cells receive chemokine stimulation. (lines 260-262 on page 11).

2) Notably, integrin activation by removal of Ca^{2+} is very rapid because the regulation of integrin affinity by Ca^{2+} does not involve a complicated intracellular signal transduction cascade. (lines 277-279 on page 12).

Reviewer #2: The manuscript by Li and colleagues investigates mechanisms by which lymphocytes are arrested at target sites along blood vessel linings and finally transmigrate through the vessel wall. Typically, this is achieved via a chemokine and integrin-mediated mechanism. In this study the authors developed assays to study integrin activation via FRET and extracellular calcium via localization of biosensors to the extracellular side of the lymphocyte plasma membrane. Based on their experimental data they put forward that chemokine or ionomycin stimulation of lymphocytes leads to decreases in extracellular calcium close to the plasma membrane that underlies integrin activation and T-cell arrest.

Point 1. My main criticism is directed to the calcium imaging data. It is implausible that decreases in extracellular calcium close to the plasma membrane could be achieved and maintained in a high calcium containing environment for the indicated periods of time (many seconds). How is this supposed to work? Could there be any extracellular matrix components that insulate the calcium space close to the membrane from the overall volume? The authors do not comment or discuss this at all. Without any further sound evidence that this could actually be true I do not believe it at all. My impression is that the authors observe some kind of artefact of the membrane-anchored indicator.

Response: We thank the reviewer for this very constructive comment. We felt sorry for that we have not provided enough data to support the conclusion that Ca^{2+} influx can induce persistent low Ca^{2+} concentration on T cell surface ($[\text{Ca}^{2+}]_{\text{ex}}$) for seconds in a high Ca^{2+} containing environment. To address the reviewer's concern that the observed $[\text{Ca}^{2+}]_{\text{ex}}$ decrease might be artefact of the membrane-anchored CEPIA, we have used an alternative approach to examine the Ca^{2+} concentration in T cell surface region. In brief, T cells were suspended in buffer containing 1.2 mM Ca^{2+} and Rhod Red (Ca^{2+} fluorescence probe). The T cell plasma membrane was visualized by FM 4-64FX (Supplementary Fig. 9). Upon ionomycin addition, Rhod Red intensity in the region close to the external plasma membrane of T cells showed a rapid decrease and reached to the lowest level in about 28

seconds (Supplementary Fig. 9a,b), indicating a rapid $[Ca^{2+}]_{ex}$ decrease on the T cell surface. These Ca^{2+} dynamics in T cell surface region showed by Rhod Red is consistent with those observed using membrane-anchored CEPIA (Fig. 3a-c). In addition, similar $[Ca^{2+}]_{ex}$ drop patterns were observed on the surface of T cells upon CCL25 treatment when using Rhod Red or membrane-anchored CEPIA (Fig. 4c and Supplementary Fig. 9c,d). Thus, Ca^{2+} influx-induced low Ca^{2+} concentration on T cell surface was consistently observed in two systems using membrane-anchored CEPIA and Ca^{2+} fluorescence probe Rhod Red, respectively. We have also used super-resolution microscopy to show that CEPIA $_{external}$ biosensors are located on the extracellular side of plasma membrane (Fig. 2d). These new data have been included in the revised manuscript (lines 305-317 on pages 13-14).

[figure redacted]

Supplementary Fig. 9: Ionomycin and CCL25 induce $[Ca^{2+}]_{ex}$ drop on T cell surface using Rhod Red indicator.

Splenic T cells isolated from *R26-LSL-CEPIA $_{external}$;Itgal-LSL-Clover;Itgb2-LSL-mRuby2;CD4-Cre* mice were suspended in buffer containing 1.2 mM Ca^{2+} and 0.6 mM Mg^{2+} and $1\times$ Rhod Red stock solution. Ionomycin (final concentration 1 μ M) or CCL25 (final concentration 0.5 μ g/ml) were added at time point 0.

a, Representative pseudocolour images of Rhod Red ratio in the solution in response to stimulation with 1 μ M ionomycin. The plasma membrane was indicated by FM 4-64FX. Scale bar, 3 μ m.

b, Time course of Rhod Red intensity on T cell surface in response to stimulation with 1 μ M ionomycin.

c, Representative pseudocolour images of Rhod Red ratio in the solution in response to stimulation with 0.5 μ g/ml CCL25. The plasma membrane was indicated by FM 4-64FX. Scale bar, 3 μ m.

d, Time course of Rhod Red intensity on T cell surface in response to stimulation with 0.5 μ g/ml CCL25.

The solid lines represent the mean; shaded areas, s.e.m. in **b** and **d**. $n = 12$ cells from 3 experiments.

Another very constructive comment from the reviewer is that the architecture of T cell surface might hinder the diffusion of Ca^{2+} from solution to the cell surface. It is known that most cells including lymphocytes are covered by a dense glycocalyx, which resides extracellularly on the cell membrane, surrounding the cell like a cloak^{3, 4}. This special

structure contributes to the formation of a physical and charged barrier modulating the flow of low molecular substrates or ions into and out of the cells^{5, 6, 7}, which may result in the delay of Ca²⁺ diffusion from the solution to the external surface of T cells. To validate this hypothesis, we have performed a new experiment to compare the diffusion rate of Ca²⁺ from solution to the surfaces of CEPIA-labeled agarose beads and T cells. The results clearly showed that addition of 1.2 mM Ca²⁺ into the solution rapidly increased [Ca²⁺]_{ex} on the surface of agarose beads (Supplementary Fig. 7). By contrast, [Ca²⁺]_{ex} on T cells surface showed slowly increase. These data indicate that Ca²⁺ diffuse slowly from solution to T cells surface. The new results have been included in the revised manuscript (lines 280-292 on pages 12-13).

Taken together, the continuous Ca²⁺-influx and slow Ca²⁺ diffusion from solution to T cell surface results in the a persistent low [Ca²⁺]_{ex} on T cells surface.

[figure redacted]

Supplementary Fig. 7: Ca²⁺ in the solution diffuses more quickly to the surface of agarose beads than to cell surface.

a, Immunoblot analysis of soluble CEPIA-HA protein with anti-HA tag antibody in culture media of 293T cells transiently transfected with CEPIA-HA or vector control.

b-c, Splenic T cells isolated from *R26-LSL-CEPIAexternal;Itgal-LSL-Clover;Itgb2-LSL-mRuby2;CD4-Cre* mice or CEPIA-HA-labeled anti-HA agarose beads were suspended in buffer containing 0 mM Ca²⁺ and 0.6 mM Mg²⁺. 1.2 mM Ca²⁺ (final concentration) were added at time point 0.

b, Representative pseudocolour images of CEPIA_{external} ratio on the surface of T cells or CEPIA-HA ratio on the surface of CEPIA-HA-labeled agarose beads in response to addition of Ca²⁺. The first two images show the distribution of CEPIA_{external} and CEPIA-HA in Em450 and Em510 channels. Scale bar, 3 μm.

c, Time course of CEPIA_{external} ratio change and the corresponding change of [Ca²⁺]_{ex} on T cell surface or changes of CEPIA-HA ratio and [Ca²⁺]_{ex} on CEPIA-HA-labeled agarose bead surface in response to addition of Ca²⁺.

The solid lines represent the mean; shaded areas, s.e.m. in **c**. *n* = 12 cells and *n* = 12 beads from 3 experiments.

Reviewer #3: Li and colleagues have studied the Ca^{2+} -dependent mechanism causing immediate rolling arrest in lymphocytes exposed to chemokines. By using an array of molecular and biophysical methods to probe $[\text{Ca}^{2+}]_{\text{ex}}$ and manipulate intracellular signals, the authors conclude that the rapid chemokine-dependent lymphocyte arrest mainly depends on a $[\text{Ca}^{2+}]_{\text{ex}}$ drop occurring within seconds, which activates $\alpha\text{L}\beta 2$ integrin and thus cell adhesion. The paper is generally interesting and the authors' suggestions original. However, to be fully convincing, the conclusion that the quick effect on cell arrest is essentially due to $[\text{Ca}^{2+}]_{\text{ex}}$ drop would require some other lines of evidence.

Point 1. My main concern is that proving the above conclusion would need more precise measurement of internal submembrane calcium, and/or more specific test of its effects. In general, it is difficult to completely block local submembrane intracellular calcium elevations merely by calcium chelators (and it is not clear from the Results if the authors have specific evidence about this in the present system). In addition, the degree of proximity between the inner mouth of the calcium pore and the inner domains of the other membrane proteins implicated in the studied effect is unknown, to the best of my knowledge. Thus (lines 193-194), SKF96365 may be more effective because, by blocking calcium influx, cuts at the root not only the $[\text{Ca}^{2+}]_{\text{ex}}$ drop but also possible local submembrane effects of calcium.

To fully demonstrate that the effect is essentially caused by external calcium depletion, one would have to apply some of the following approaches: a) attempting to measure local internal calcium changes as the authors did for the external; b) block/inactivate/delete the possible submembrane targets of rapid calcium effects (are these targets known?); d) have the authors ever attempted to release caged calcium intracellularly and test the kinetics of leukocyte arrest? e) would it be possible to carry out complementary experiments, in the presence of increased $[\text{Ca}^{2+}]_{\text{ex}}$?

Response: We thank the reviewer for the very constructive comment and suggestions. As suggested, we have performed new experiment to measure the local internal submembrane calcium ($[Ca^{2+}]_{in}$) changes as we did for the external. We established a plasma membrane-anchored internal CEPIA (*CEPIA_{internal}*) by adding PDGFR-TM sequence to the N-terminus of CEPIA (Supplementary Fig. 10a). By using super-resolution microscopy, we confirmed that *CEPIA_{internal}* was located at the internal side of cell plasma membrane (Supplementary Fig. 10b). CCL25 treatment induced a rapid increase in *CEPIA_{internal}* ratio and $[Ca^{2+}]_{in}$ in the first few seconds, then recovered back gradually (Supplementary Fig. 10c,d), which is similar to the results of CCL25-induced $[Ca^{2+}]_{cyto}$ change (Supplementary Fig. 4). These new results indicate that CCL25-triggered Ca^{2+} influx can induce similar Ca^{2+} increase patterns in both internal side of cell plasma membrane and cytosol.

Furthermore, blocking Ca^{2+} influx with SKF96365 not only inhibited $[Ca^{2+}]_{cyto}$ and $[Ca^{2+}]_{in}$ increases as did by chelating intracellular Ca^{2+} with BAPTA-AM (Supplementary Fig. 4 and Supplementary Fig. 10d), but also blocked CCL25-induced $[Ca^{2+}]_{ex}$ drop (Fig. 4f). The major difference between the effects of SKF96365 and BAPTA-AM is that SKF96365 but not BAPTA-AM can inhibit CCL25-induced $[Ca^{2+}]_{ex}$ drop (Fig. 4f and Supplementary Fig. 10e) and its associated $\alpha L\beta 2$ quick activation in 28 second upon chemokine stimulation (Fig. 4g,e), suggesting $\alpha L\beta 2$ quick activation is essentially caused by Ca^{2+} influx-induced $[Ca^{2+}]_{ex}$ drop.

These new results and discussion have been included in the revised manuscript (lines 318-334 on page 14).

[figure redacted]

Supplementary Fig. 10: Establishing a cell membrane-anchored *CEPIA_{internal}* to monitor intracellular submembrane Ca^{2+} ($[Ca^{2+}]_{in}$) dynamics.

a, Schematic diagram of the experimental setup for the membrane-anchored *CEPIA_{internal}*.

b-d, Splenic T cells from WT mice transfected with *CEPIA_{internal}* were suspended in buffer containing 1.2 mM Ca^{2+} and 0.6 mM Mg^{2+} . CCL25 (final concentration 0.5 $\mu g/ml$) were added at time point 0.

b, Representative fluorescence images of T cells showing the distribution of CEPIA_{internal}. The plasma membrane as indicated by FM 4-64FX. Scale bar, 3 μm .

c, Representative pseudocolour images of CEPIA_{internal} ratio in response to stimulation with 0.5 $\mu\text{g/ml}$ CCL25. The first two images show the distribution of CEPIA_{internal} in Em450 and Em510 channels. Scale bar, 3 μm .

d, Time course of CEPIA_{internal} ratio change and the corresponding change of $[\text{Ca}^{2+}]_{\text{in}}$ in T cells pretreated with 100 μM BAPTA-AM, 100 μM SKF96365 or vehicle control in response to stimulation with 0.5 $\mu\text{g/ml}$ CCL25.

e, Splenic T cells isolated from *R26-LSL-CEPIAexternal;Itgal-LSL-Clover;Itgb2-LSL-mRuby2;CD4-Cre* mice were suspended in buffer containing 1.2 mM Ca^{2+} and 0.6 mM Mg^{2+} . CCL25 (final concentration 0.5 $\mu\text{g/ml}$) were added at time point 0. Time course of CEPIA_{external} ratio change and the corresponding change of $[\text{Ca}^{2+}]_{\text{ex}}$ in T cells pretreated with 100 μM BAPTA-AM, 100 μM SKF96365 or vehicle control in response to stimulation with 0.5 $\mu\text{g/ml}$ CCL25.

The solid lines represent the mean; shaded areas, s.e.m in **d** and **e**. $n = 12$ cells from 3 experiments.

[figure redacted]

Supplementary Fig. 4: Cytosolic Ca^{2+} ($[\text{Ca}^{2+}]_{\text{cyto}}$) change in mouse splenic T cells pretreated with 100 μM BAPTA-AM, 100 μM SKF96365 or vehicle control in response to stimulation with 0.5 $\mu\text{g/ml}$ CCL25 was detected by Fluo-4.

Splenic T cells were isolated from *R26-LSL-CEPIAexternal;Itgal-LSL-Clover;Itgb2-LSL-mRuby2;CD4-Cre* mice and suspended in buffer containing 1.2 mM Ca^{2+} and 0.6 mM Mg^{2+} . CCL25 (final concentration 0.5 $\mu\text{g/ml}$) were added at time point 0.

The solid lines represent the mean; shaded areas, s.e.m. $n = 30$ cells from 3 experiments.

Point 2. Based on the results with SKF96365, do the authors think that the quick chemokine-induced calcium influx essentially depends on store-operated calcium entry (SOCE) activation? This is not very clear in the text. If so, is the known kinetics of SOCE activation (TRP, Ora-1?) fully consistent with the quick time course of extracellular calcium depletion?

Response: We thank the reviewer for this very constructive comment. Chemokine-induced calcium influx has traditionally been thought to be achieved through the activation of store-operated calcium entry (SOCE), a molecular complex primarily composed of STIM1 and ORAI1, of which ORAI1 is identified as part of the Ca^{2+} release-activated Ca^{2+} (CRAC) channel pore^{8, 9}. SOCE assembly requires the recruitment of STIM1 from the ER to proximity with ORAI1 at the cell membrane, leading to a dynamic complex formation

typically occurring within seconds¹⁰. Several studies have reported the rapid orchestration of SOCE complex formation coinciding with the dynamic reduction of extracellular Ca^{2+} in a similar time scale as we described in our work^{11, 12, 13}. Thus, the kinetics of SOCE activation is consistent with the quick time course of chemokine-induced cell surface Ca^{2+} drop.

We have described this point in the Discussion section in the revised manuscript (lines 340-345 on page 15).

Point 3. Fig. 1 and 2. Does nominal 0 Ca^{2+} really correspond to $[\text{Ca}^{2+}]_o = 0$? Calcium contaminations are known to be present in salts and water (in the order of 0.1 mM). Did the authors ever tested EGTA as in panel F? The ratio observed in the presence of EGTA (approximately 0.85) does not correspond to the ratio observed in 0 calcium in panel E. Could the flat part of the CEPIA ratio below 0.1 mM partly artefactual? Perhaps a control with EGTA would be also useful here.

Response: We thank the reviewer for this very constructive comment. We agree with the reviewer that 0 mM $[\text{Ca}^{2+}]$ in Fig. 1a,b,d,e and Fig. 2e,f was not absolutely zero because of the trace calcium in the buffer (KCl/MOPS) that we used in these experiments. We have analyzed the concentration of Ca^{2+} in the buffer with Amplite™ Colorimetric Calcium Quantitation Kit (36361, AAT Bioquest) which has a limit of detection at 2.34 μM Ca^{2+} . The results showed that Ca^{2+} was undetectable in this buffer, indicating a $[\text{Ca}^{2+}]$ below 2.34 μM . Thus, we have added a note in the revised Fig. 1 legend by stating that 0 mM $[\text{Ca}^{2+}]$ means a Ca^{2+} concentration below the detection limit (2.34 μM Ca^{2+}) of the Calcium Quantitation Kit (36361, AAT Bioquest) (lines 693-694 on page 29).

As for the EGTA treatment results in Fig. 1f, we have repeated the experiment with a new batch of EGTA reagent (MedChemExpress, CAS No. 67-42-5, Cat. No. HY-D0861). The results showed that $F_{\text{mRuby2}}/F_{\text{Clover}}$ ratio decreased immediately within seconds upon

EGTA treatment, and finally stabilized at the level of approximately 0.7, which corresponded to the values of 0.05 mM to 0 mM Ca²⁺ in Fig. 1e.

These revision and new data have been updated in the revised manuscript (lines 407-408 on page 18).

[figure redacted]

Fig. 1: e, Quantification of α L β 2 tail FRET ratio. The FRET ratio of each cell was normalized to the mean value of cells in 1.2 mM Ca²⁺. Data are shown as box-and-whisker plots showing the range, minimum, maximum and mean ($n = 30$ cells for each condition from 3 experiments).
f, Time course of α L β 2 tail FRET ratio change in T cells on the immobilized ICAM-1 (20 μ g/ml) substrates upon chelation of Ca²⁺ with 5 mM EGTA in buffer containing 1.2 mM Ca²⁺ plus 0.6 mM Mg²⁺ (left). EGTA was added at time point 0. The FRET ratio change was normalized to the mean value of cells before EGTA treatment. The solid lines represent the mean; shaded areas, s.e.m. ($n = 50$ cells from 3 experiments). The statistic results at representative time points were shown (right). Data represent the mean \pm s.e.m. in **f**. ** $P < 0.01$; **** $P < 0.0001$; ns, not significant (one-way ANOVA with Dunnett's test in **e** to compare the means of different Ca²⁺ concentration groups to the mean of 1.2 mM Ca²⁺ group; unpaired two-tailed Student's t -test in **f** to compare the means of 15 s and 30 s groups to the mean of 0 s group).

Point 4. Discussion. The first part of the paper is generally clearly written, but the writing style deteriorates considerably in the Discussion. In general, this section seems somewhat perfunctory.

Response: Thanks for this very constructive comment. We have revised the Discussion section extensively in the revised manuscript.

Point 5. Introduction/p. 5/elsewhere. It should be perhaps made clear at the outset that the authors are referring to free calcium and magnesium, not total blood concentration. In fact, the titles of the cited references (16-17) mention 'serum' not blood.

Response: Thanks for the comments. The contents have been updated in the revised manuscript as shown below.

“Integrins are metalloproteins and their functions are strictly dependent on and regulated by free Ca²⁺ and Mg²⁺ that physiologically exist in serum at millimolar-level”.

(lines 59-60 on page 3).

“The physiological concentrations of free Ca^{2+} and Mg^{2+} in serum (1.2 mM Ca^{2+} and 0.6 mM Mg^{2+}) were set as control”. (lines 90-92 on page 5).

Point 6. Line 88. ICAM-1 is wrongly defined as ‘intracellular adhesion molecule-1’ (instead of ‘intercellular...etc.’), which could make the paragraph puzzling to the general reader. Moreover, the experimental system should be briefly explained (ICAM-1 was immobilized on culture dishes), as the Fig. 1 legend is also not very clear.

Response: We felt sorry for this mistake. We have corrected this mistake with the right term: intercellular cell adhesion molecule-1 (ICAM-1) (line 89 on page 4). As suggested, we have also clearly described the experimental setup in the Fig. 1 legend as following.

“ICAM-1 (20 $\mu\text{g}/\text{ml}$) was immobilized on petri dishes. Adhesion of T cells to the immobilized ICAM-1 substrates at a wall shear stress of 1 dyn/cm^2 was examined ($n = 6$)”. (lines 695-697 on page 29).

Point 7. Line 139. ‘western’ or ‘Western’?

Response: Thanks for the comments. This word has been updated in the revised manuscript (line 140 on page 7).

Point 8. Line 171. ‘BAMPTA’

Response: Sorry for this typo. The mistake has been corrected in the revised manuscript (line 176 on page 8).

Point 9. p. 9 Lines 180 (...was not affected..) and 183 (...partially prevented...) seem somewhat contradictory. The entire paragraph could be reworded for clarity.

Response: We felt sorry for not describing this point clearly. Conventional **slow integrin inside-out activation** by chemokines includes intracellular Ca^{2+} -induced and intracellular

Ca²⁺-independent activation of integrins¹², which is distinct from the **quick integrin activation** triggered by [Ca²⁺]_{ex} drop on T cell surface. In our experiment, the conventional inside-out activation of αLβ2 by CCL25 emerged **after 28 seconds** upon chemokine stimulation. Chelating intracellular Ca²⁺ with BAPTA-AM (Supplementary Fig. 4) partially prevented the decrease in integrin tail FRET signal **after 28 seconds** upon CCL25 treatment (Fig. 4e), suggesting BAPTA-AM inhibited the intracellular Ca²⁺-induced slow integrin inside-out activation but did not affect intracellular Ca²⁺-independent integrin inside-out activation. In contrast, chelating intracellular Ca²⁺ with BAPTA-AM treatment did not affect the quick αLβ2 activation associated with [Ca²⁺]_{ex} drop on T cell surface **in 28 seconds** upon CCL25 stimulation (Fig. 4d,e), indicating it is distinct from the conventional slow integrin inside-out activation.

As suggested, we have rewritten this paragraph in the revised manuscript as shown below (lines 183-195 on pages 8-9).

“In line with these results, CCL25 treatment induced a rapid decrease in integrin tail FRET signal in 28 seconds, suggesting a quick activation of αLβ2 that was triggered by the decrease in Ca²⁺ on T cell surface. During this period, αLβ2 activation was not affected by chelating intracellular Ca²⁺ with BAPTA-AM treatment (Fig. 4d,e), indicating it is distinct from the conventional slow integrin inside-out activation.

Conventional slow integrin inside-out activation by chemokines includes intracellular Ca²⁺-induced and intracellular Ca²⁺-independent activation of integrins¹². Chelating intracellular Ca²⁺ with BAPTA-AM (Supplementary Fig. 4) partially prevented the decrease in integrin tail FRET signal after 28 seconds upon CCL25 treatment (Fig. 4e), suggesting BAPTA-AM inhibited the intracellular Ca²⁺-induced slow integrin inside-out activation but did not affect intracellular Ca²⁺-independent integrin inside-out activation”.

Point 10. Line 187. The action and specificity of SKF96365 should be explained here.

Response: As suggested, we have explained the action and specificity of SKF96365 in the revised manuscript (lines 195-196 on page 9).

“SKF96365 is a store-operated calcium entry (SOCE) inhibitor and also blocks TRPC channels and voltage-gated Ca^{2+} channels^{14, 15}”.

Point 11. Line 325. Because dyn/cm^2 are the physical dimensions of pressure and not flow rate, this should be perhaps briefly explained.

Response: Thanks for the comments. Shear stress (dyn/cm^2) is generally used in flow chamber assay to reflect the force on the lower surface of chamber where cells form adhesion to substrates^{16, 17}. We should use the term “shear stress” instead of “flow rate” here. Thus, we have revised this sentence in the revised manuscript as below (lines 384-387 on page 17).

“Cells were then diluted to 1×10^6 cells/ml in HBSS containing 0.6 mM Mg^{2+} and different concentrations of Ca^{2+} ranging from 0 to 1.2 mM, and immediately perfused through the flow chamber at a constant **shear stress of 1 dyn/cm^2** for 1 min”.

Point 12. Statistics. In Fig. 1, it is not clear which pairs were compared with the post-hoc test.

Response: Thanks for the comments. We have marked the pairs for comparison in the legend for Fig. 1 (lines 718-721 on page 30).

“one-way ANOVA with Dunnett’s test in **a**, **b** and **e** to compare the means of different Ca^{2+} concentration groups to the mean of 1.2 mM Ca^{2+} group; unpaired two-tailed Student’s *t*-test in **f** to compare the means of 15 s and 30 s groups to the mean of 0 s group”.

References:

1. Groenendyk J, Agellon LB, Michalak M. Calcium signaling and endoplasmic reticulum stress. *Int Rev Cell Mol Biol* **363**, 1-20 (2021).
2. Suzuki J, Kanemaru K, Ishii K, Ohkura M, Okubo Y, Iino M. Imaging intraorganellar Ca²⁺ at subcellular resolution using CEPIA. *Nature Communications* **5**, (2014).
3. Mockl L. The Emerging Role of the Mammalian Glycocalyx in Functional Membrane Organization and Immune System Regulation. *Front Cell Dev Biol* **8**, 253 (2020).
4. Springer TA. Adhesion receptors of the immune system. *Nature* **346**, 425-434 (1990).
5. Yang R, Chen M, Zheng J, Li X, Zhang X. The Role of Heparin and Glycocalyx in Blood-Brain Barrier Dysfunction. *Front Immunol* **12**, 754141 (2021).
6. Sauvanet C, Wayt J, Pelaseyed T, Bretscher A. Structure, regulation, and functional diversity of microvilli on the apical domain of epithelial cells. *Annu Rev Cell Dev Biol* **31**, 593-621 (2015).
7. Kutuzov N, Flyvbjerg H, Lauritzen M. Contributions of the glycocalyx, endothelium, and extravascular compartment to the blood-brain barrier. *Proc Natl Acad Sci U S A* **115**, E9429-E9438 (2018).
8. Trebak M, Kinet JP. Calcium signalling in T cells. *Nat Rev Immunol* **19**, 154-169 (2019).
9. Emrich SM, Yoast RE, Trebak M. Physiological Functions of CRAC Channels. *Annu Rev Physiol* **84**, 355-379 (2022).
10. Feske S. Calcium signalling in lymphocyte activation and disease. *Nat Rev Immunol* **7**, 690-702 (2007).
11. Shaw PJ, Feske S. Physiological and pathophysiological functions of SOCE in the immune system. *Front Biosci (Elite Ed)* **4**, 2253-2268 (2012).
12. Dixit N, Simon SI. Chemokines, selectins and intracellular calcium flux: temporal and spatial cues for leukocyte arrest. *Frontiers in Immunology* **3**, (2012).
13. Vaeth M, Kahlfuss S, Feske S. CRAC Channels and Calcium Signaling in T Cell-Mediated Immunity. *Trends Immunol* **41**, 878-901 (2020).
14. Singh A, Hildebrand ME, Garcia E, Snutch TP. The transient receptor potential channel antagonist SKF96365 is a potent blocker of low-voltage-activated T-type calcium channels. *Br J Pharmacol* **160**, 1464-1475 (2010).
15. Song M, Chen D, Yu SP. The TRPC channel blocker SKF 96365 inhibits glioblastoma cell growth by enhancing reverse mode of the Na⁽⁺⁾ /Ca⁽²⁺⁾ exchanger and increasing intracellular Ca⁽²⁺⁾. *Br J Pharmacol* **171**, 3432-3447 (2014).
16. Chen J, Salas A, Springer TA. Bistable regulation of integrin adhesiveness by a bipolar metal ion cluster. *Nat Struct Biol* **10**, 995-1001 (2003).
17. Lin C, *et al.* Fever Promotes T Lymphocyte Trafficking via a Thermal Sensory Pathway Involving Heat Shock Protein 90 and alpha4 Integrins. *Immunity* **50**, 137-151 e136 (2019).

REVIEWER COMMENTS

Reviewer #1 (Remarks to the Author):

I appreciate the efforts made by the authors to address the reviewers' concerns and revise the manuscript accordingly. Overall, I am satisfied with the revised version and find no major issues with the scientific content of the paper.

However, I have some reservations regarding the presentation of the new results in the manuscript. While the authors have addressed the reviewers' concerns by including supplementary figures 7 to 10 in the discussion, I believe that this approach is not conventional. Placing supplementary data in the discussion section, especially without proper integration into the main narrative, could make it challenging for readers to follow the flow of information.

Instead, I would suggest that the authors incorporate these supplementary findings into the main body of the paper. Even if some results are not entirely conclusive or remain somewhat speculative, they may be relevant to the main events as they are being revealed in the manuscript. For instance, Supplementary Fig. 8, which explores the potential reason for the disappearance of intracellular calcium into the ER, could be integrated into the main text to provide context for that specific problem as it appears in the text.

Also, through this simplification, the authors could avoid making the discussion overly technical and extensive. This approach would ensure that readers have a more coherent and comprehensive understanding of the research findings without having to navigate through supplementary materials separately.

Reviewer #2 (Remarks to the Author):

The revised manuscript by Li and colleagues now presents new supporting data to show that extracellular calcium close to the plasma membrane may indeed be reduced for periods of up to minutes in spite of solution calcium remaining high (over 1 mM). Data in supplementary figures 7 and 9 support the idea that an extracellular matrix component or

glycocalix may help in insulating submembrane outer calcium domains from solution calcium, thus my initial concerns are mitigated. There also seems to be no artefact of the CEPIA indicator localized to the outside of the membrane. On the other hand, many other exogenous compounds seem to be able to access this submembrane space without apparent problems. Thus, many questions on the role of this glycocalix on calcium handling remain unclear to me. I am aware that addressing these issues may be out of the scope of the current manuscript, so I consider this an editorial decision. But I have to state that I am still uneasy about the general finding.

Reviewer #3 (Remarks to the Author):

The revised version addresses most of my previous concerns.

I notice a couple of relatively minor issues:

- 1) There are now supplementary data obtained with the new probe CEPIAinternal. Should not all this be part of the Results section instead of the Discussion?
- 2) As is also stated by the authors (lines 195-196), SKF96365 blocks all classes of CaV, at the high concentration used (100 μ M). Did the authors choose such concentration to block all the main calcium entry pathways? Or do they nonetheless attribute most of the chemokine effect to SOCE? This point is still not very clear in the revised version.
- 3) Statistical analysis: please state the normality and variance homogeneity tests used before applying the statistical parametric methods.

Re: NCOMMS-23-42688B

Response to reviewers' comments

We thank all reviewers for the review of our revised manuscript. We are glad that most of the reviewers are satisfied with our revisions. As to the remaining issues raised by the reviewers, we have addressed all of them with additional experiment, better interpretation of our data and discussion of related literatures, which should help to further improve the overall quality of the manuscript. All of the revisions have been highlighted in red in the revised manuscript. The point-by-point response is shown as below.

Reviewer #1: I appreciate the efforts made by the authors to address the reviewers' concerns and revise the manuscript accordingly. Overall, I am satisfied with the revised version and find no major issues with the scientific content of the paper.

However, I have some reservations regarding the presentation of the new results in the manuscript. While the authors have addressed the reviewers' concerns by including supplementary figures 7 to 10 in the discussion, I believe that this approach is not conventional. Placing supplementary data in the discussion section, especially without proper integration into the main narrative, could make it challenging for readers to follow the flow of information.

Instead, I would suggest that the authors incorporate these supplementary findings into the main body of the paper. Even if some results are not entirely conclusive or remain somewhat speculative, they may be relevant to the main events as they are being revealed in the manuscript. For instance, Supplementary Fig. 8, which explores the potential reason for the disappearance of intracellular calcium into the ER, could be integrated into the main text to provide context for that specific problem as it appears in the text.

Also, through this simplification, the authors could avoid making the discussion overly technical and extensive. This approach would ensure that readers have a more coherent and comprehensive understanding of the research findings without having to navigate through supplementary materials separately.

Response: We sincerely appreciate your satisfaction with our revisions and the very constructive suggestions regarding the presentation of the new results. As suggested, we have made the following changes in the revised manuscript.

- 1) We have changed Supplementary Fig. 8 to Fig. 4 and moved the data to the results section (please refer to lines 184-196 on pages 8-9).
- 2) We have changed Supplementary Fig. 7 to Supplementary Fig. 5, Supplementary Fig. 9 to Supplementary Fig. 4, and Supplementary Fig. 10 to Supplementary Fig. 7 and moved these data to the main narrative in the revised manuscript (please refer to lines 161-168 on pages 7-8, lines 197-207 on page 9, lines 215-217 on page 10 and lines 239-256 on page 11).

Reviewer #2: The revised manuscript by Li and colleagues now presents new supporting data to show that extracellular calcium close to the plasma membrane may indeed be reduced for periods of up to minutes in spite of solution calcium remaining high (over 1 mM). Data in supplementary figures 7 and 9 support the idea that an extracellular matrix component or glycocalyx may help in insulating submembrane outer calcium domains from solution calcium, thus my initial concerns are mitigated. There also seems to be no artefact of the CEPIA indicator localized to the outside of the membrane. On the other hand, many other exogenous compounds seem to be able to access this submembrane space without apparent problems. Thus, many questions on the role of this glycocalyx on calcium handling remain unclear to me. I am aware that addressing these issues may be out of the scope of the current manuscript, so I consider this an editorial decision. But I have to state that I am still uneasy about the general finding.

Response: We are glad that our revisions mitigated the reviewer's concerns. As to the role of T cell glycocalyx on hindering Ca^{2+} diffusion, in addition to adding a more comprehensive discussion on this topic (lines 316-339 on pages 14-15, also shown as below), we have also performed a new experiment to clarify the role of the glycocalyx on calcium handling. In brief, we treated the T cells with PNGase F¹ to remove all types of N-glycans from T cell surface and analyzed ionomycin-induced extracellular Ca^{2+} decrease using Rhod Red indicator. We did not use CEPIA sensor in this experiment because glycation of calmodulin (CaM), the core component of CEPIA sensor, is associated with its activation capacity and Ca^{2+} binding capacity², and PNGase F treatment may disrupt the normal function of CEPIA. The results clearly showed that removal of N-glycans on T cell surface by PNGase F treatment significantly attenuated ionomycin-induced $[\text{Ca}^{2+}]_{\text{ex}}$ decrease (Supplementary Fig. 4e,f), suggesting the reduced hinder of Ca^{2+} diffusion from solution to T cell surface after removing cell surface glycocalyx. Thus, glycocalyx structures can hinder Ca^{2+} diffusion from solution to T cell surface (please refer to lines 336-339 on pages 14-15).

[figure redacted]

Supplementary Fig. 4: Splenic T cells isolated from WT mice were suspended in buffer containing 1.2 mM Ca^{2+} and 0.6 mM Mg^{2+} and 1× Rhod Red stock solution. Ionomycin (final concentration 1

μM) was added at time point 0. T cells were pretreated with 1000 units/ml PNGase F in e and f. e, Representative pseudocolour images of Rhod Red ratio in the solution in response to stimulation with 1 μM ionomycin. The plasma membrane was indicated by FM 4-64FX. Scale bar, 3 μm . f, Time course of Rhod Red intensity in PNGase F-treated T cell surface region in response to stimulation with 1 μM ionomycin.

The solid lines represent the mean; shaded areas, s.e.m. in f. $n = 12$ cells from 3 experiments.

The discussion on the role of cell surface glycocalyx (lines 316-339 on pages 14-15 in the revised manuscript):

“All living cells have extracellular polysaccharide structures attached. Glycocalyx is the universal term that describes this structure, which is a dynamic surface layer composed of proteoglycans, glycoproteins, and glycosaminoglycans³. Glycocalyx is a dense, gel-like meshwork that surrounds the cell and fulfills a multiplicity of functions to cells, including creating a physical and chemical barrier, allowing for buffering extracellular compounds⁴. Especially, the negatively charged barrier of the glycocalyx impedes the flow of cationic molecules between the cell and its surrounding environment, in particular the divalent cation calcium because of its charge density in the plasma space⁵. Lymphocyte glycocalyx components such as CD45, transmembrane tyrosine phosphatase CD148, transmembrane mucins, and the pericellular matrix generated by hyaluronan and CD44, could shield short receptors through its long and rigid glycocalyx structures⁶. These large glycoproteins on T lymphocytes, presenting a steric barrier perpendicular to the plasma membrane, constrains its lateral diffusion and forms a barrier of varying density and length that can curtail access of macromolecules and particulate material to the surface of the cell^{7, 8}. Several studies have demonstrated that degradation of glycocalyx using endogenous heparinase, hyaluronidase or neutralizing the negative charge of the glycocalyx by myeloperoxidase can facilitate water and low-density lipoprotein transport thus enhancing cell permeability^{9, 10, 11}, indicating its crucial role in maintaining the permeable barrier of microcirculation. Indeed, our data showed that Ca^{2+} diffused slowly from solution to T cell surface. Removal of N-glycans on T cell surface by PNGase F treatment¹ significantly attenuated ionomycin-induced $[\text{Ca}^{2+}]_{\text{ex}}$ decrease (Supplementary Fig. 4e,f), suggesting the reduced hinder of Ca^{2+} diffusion from solution to T cell surface after removing cell surface glycocalyx.”

Reviewer #3: The revised version addresses most of my previous concerns. I notice a couple of relatively minor issues:

Point 1. There are now supplementary data obtained with the new probe CEPIA_{internal}. Should not all this be part of the Results section instead of the Discussion?

Response: We sincerely appreciate your satisfaction with the revised manuscript and the very constructive suggestions regarding the presentation of the new results. As suggested, we have made the following changes to move some data from discussion to main text in the revised manuscript.

- 1) We have changed Supplementary Fig. 8 to Fig. 4 and moved the data to the results section (please refer to lines 184-196 on pages 8-9).
- 2) We have changed Supplementary Fig. 7 to Supplementary Fig. 5, Supplementary Fig. 9 to Supplementary Fig. 4, and Supplementary Fig. 10 to Supplementary Fig. 7 and moved these data to the main narrative in the revised manuscript (please refer to lines 161-168 on pages 7-8, lines 197-207 on page 9, lines 215-217 on page 10 and lines 239-256 on page 11).

Point 2. As is also stated by the authors (lines 195-196), SKF96365 blocks all classes of Ca_v, at the high concentration used (100 μM). Did the authors choose such concentration to block all the main calcium entry pathways? Or do they nonetheless attribute most of the chemokine effect to SOCE? This point is still not very clear in the revised version.

Response: We thank the reviewer for the comment. SKF96365 is a store-operated calcium entry (SOCE) inhibitor and also blocks TRPC channels and voltage-gated Ca²⁺ channels^{12, 13}. In this study, we chose the concentration of 100 μM to block all the main calcium entry pathways. We have made this point clear in the revised manuscript (line 431 on page 19 and line 529 on page 23).

In addition, CCL25 can induce rapid Ca²⁺ influx through SOCE and CRAC channels when binds to its receptor CCR9^{14, 15}. Our data showed that SKF96365 can potently obstruct CCL25-induced Ca²⁺-influx (Fig. 5f and Supplementary Fig. 6). We have clearly

clarified this point in the revised manuscript (please refer to lines 209-211 on page 9 and 229-231 on page 10).

Point 3. Statistical analysis: please state the normality and variance homogeneity tests used before applying the statistical parametric methods.

Response: We thank the reviewer for this very constructive comment. We have stated the normality and variance homogeneity tests used before applying the statistical parametric methods in the revised manuscript (please refer to lines 553-560 on page 24, lines 765-768 on page 32, lines 822-823 on page 34, lines 874-875 on page 36 and lines 890-891 on page 37).

References:

1. Tang F, *et al.* Selective N-glycan editing on living cell surfaces to probe glycoconjugate function. *Nat Chem Biol* **16**, 766-775 (2020).
2. Kowluru RA, *et al.* Glycation of calmodulin: chemistry and structural and functional consequences. *Biochemistry* **28**, 2220-2228 (1989).
3. Mortazavi CM, Hoyt JM, Patel A, Chignalia AZ. The glycocalyx and calcium dynamics in endothelial cells. *Curr Top Membr* **91**, 21-41 (2023).
4. Mockl L. The Emerging Role of the Mammalian Glycocalyx in Functional Membrane Organization and Immune System Regulation. *Front Cell Dev Biol* **8**, 253 (2020).
5. Foote CA, *et al.* Endothelial Glycocalyx. *Compr Physiol* **12**, 3781-3811 (2022).
6. Springer TA. Adhesion receptors of the immune system. *Nature* **346**, 425-434 (1990).
7. Ostrowski PP, Grinstein S, Freeman SA. Diffusion Barriers, Mechanical Forces, and the Biophysics of Phagocytosis. *Dev Cell* **38**, 135-146 (2016).
8. Kuo JC, Paszek MJ. Glycocalyx Curving the Membrane: Forces Emerging from the Cell Exterior. *Annu Rev Cell Dev Biol* **37**, 257-283 (2021).
9. Dull RO, Hahn RG. The glycocalyx as a permeability barrier: basic science and clinical evidence. *Crit Care* **26**, 273 (2022).
10. Jin J, *et al.* The Structure and Function of the Glycocalyx and Its Connection With Blood-Brain Barrier. *Front Cell Neurosci* **15**, 739699 (2021).
11. Gao L, Lipowsky HH. Composition of the endothelial glycocalyx and its relation to its thickness and diffusion of small solutes. *Microvasc Res* **80**, 394-401 (2010).
12. Singh A, Hildebrand ME, Garcia E, Snutch TP. The transient receptor potential channel antagonist SKF96365 is a potent blocker of low-voltage-activated T-type calcium channels. *Br J Pharmacol* **160**, 1464-1475 (2010).
13. Song M, Chen D, Yu SP. The TRPC channel blocker SKF 96365 inhibits glioblastoma cell growth by enhancing reverse mode of the Na(+)/Ca(2+) exchanger and increasing intracellular Ca(2+). *Br J Pharmacol* **171**, 3432-3447 (2014).
14. Agace WW. Tissue-tropic effector T cells: generation and targeting opportunities. *Nature Reviews Immunology* **6**, 682-692 (2006).
15. Feske S. Calcium signalling in lymphocyte activation and disease. *Nat Rev Immunol* **7**, 690-702 (2007).

REVIEWERS' COMMENTS

Reviewer #2 (Remarks to the Author):

The additional experiment with PNGase F in new supplementary Figure 4 help to dissipate my concerns somewhat. I am now ok that this work should be published. The data of supplementary fig. 4 could actually become part of a main figure.

Reviewer #3 (Remarks to the Author):

Thanks for the revision. I have no further comments.

Re: NCOMMS-23-42688C

Response to reviewers' comments

We thank all reviewers for the review of our revised manuscript. We are glad that the reviewers are satisfied with our revisions. As to the remaining issue raised by the reviewer, we have addressed this in the revised manuscript, which should help to further improve the overall quality of the manuscript. The point-by-point response is shown as below.

Reviewer #2: The additional experiment with PNGase F in new supplementary Figure 4 help to dissipate my concerns somewhat. I am now ok that this work should be published. The data of supplementary fig. 4 could actually become part of a main figure.

Response: We sincerely appreciate your satisfaction with our revisions and the very constructive suggestions regarding the presentation of the new results. As suggested, we have moved Supplementary Fig. 4 to Fig. 6 as one of the main figures in the revised manuscript.